# Lateral meltwater transfer across an Antarctic ice shelf

**Rebecca Dell[1,2], Neil Arnold[1], Ian Willis[1], Alison Banwell[3,1], Andrew Williamson[1], Hamish Pritchard[2], and Andrew Orr[2]**

[1]Scott Polar Research Institute, Lensfield Road, Cambridge, CB2 1ER, UK

[2]British Antarctic Survey, High Cross, Madingley Road, Cambridge, CB3 0ET, UK

[3]Cooperative Institute for Research in Environmental Sciences, University of Colorado Boulder, Boulder, CO, 80309, USA

copernicus-publications

*Correspondence to: Rebecca Dell (rld46@cam.ac.uk)*

## Abstract

Surface meltwater on ice shelves can exist as slush, it can pond in lakes or crevasses, or it can flow in surface streams and rivers. The collapse of the Larsen B Ice Shelf in 2002 has been attributed to the sudden drainage of ~3000 surface lakes, and has highlighted the potential for surface water to cause ice-shelf instability. Surface meltwater systems have been identified across numerous Antarctic ice shelves, although the extent to which these systems impact ice-shelf instability is poorly constrained. To better understand the role of surface meltwater systems on ice shelves, it is important to track their seasonal development, monitoring the fluctuations in surface water volume and the transfer of water across ice-shelf surfaces. Here, we use Landsat 8 and Sentinel-2 imagery to track surface meltwater across the Nivlisen Ice Shelf in the 2016-2017 melt season. We develop the Fully Automated Supraglacial-Water Tracking algorithm for Ice Shelves (FASTISh) and use it to identify and track the development of 1598 water bodies, which we classify as either circular or linear. The total volume of surface meltwater peaks on 26th January 2017 at $5.5 \times 10^7 \, m^3$. At this time, 63% of the total volume is held within two linear surface meltwater systems, which are up to 27 km long, are orientated along the ice shelf's north-south axis, and follow the surface slope. Over the course of the melt season, they appear to migrate away from the grounding line, while growing in size and enveloping smaller water bodies. This suggests there is large-scale lateral water transfer through the surface meltwater system and the firn pack towards the ice-shelf front during the summer.

## 1 Introduction

The total mass loss from Antarctica has increased from $40 \pm 9$ Gt/y in 1979–1990 to $252 \pm 26$ Gt/y in 2009–2017, providing a cumulative contribution to sea-level rise of $14.0 \pm 2.0$ mm since 1979 (Rignot et al., 2019). Mass loss from Antarctica will likely increase in the near future due, at least in part, to the shrinkage and thinning of some of its ice shelves (Kuipers Munneke et al., 2014; DeConto and Pollard, 2016; Siegert et al., 2019) and the associated acceleration of inland ice across the grounding lines (Fürst et al., 2016; Gudmundsson et al., 2019). Seven out of 12 ice-shelves that bordered the Antarctic Peninsula have collapsed in the last 50 years (Cook and Vaughan, 2010). One of the most notable events was the

February-March 2002 collapse of Larsen B, leading to both an instantaneous and a longer term speedup of the glaciers previously buttressed by the ice shelf (Scambos et al., 2004; Wuite et al., 2015; De Rydt et al., 2015), and resulting in their increased contribution to sea-level rise (Rignot et al., 2004).

The unforeseen catastrophic disintegration of Larsen B highlighted the unpredictable nature of ice-shelf collapse, and prompted a search for the causes of ice-shelf instability. Current understanding of the factors causing ice-shelf instability stems from the very limited number of airborne and satellite observations prior to and following collapse events (e.g. Glasser and Scambos, 2008; Scambos et al. 2009; Banwell et al., 2014, Leeson et al., 2020), numerical modelling (e.g. Vieli et al. 2006, Banwell et al. 2013, Banwell and MacAyeal, 2015), and the few in-situ measurements investigating recent and current ice-shelf processes (e.g. Hubbard et al. 2016; Bevan et al., 2017; Banwell et al. 2019). It has been suggested that the chain reaction drainage of ~3000 surface meltwater lakes, which covered 5.3% of the total ice-shelf area and had a mean depth of 0.82 m (Banwell et al., 2014), may have triggered the near-instantaneous break-up of Larsen B (Banwell et al., 2013; Robel and Banwell, 2019), highlighting the potential importance of surface hydrology for ice-shelf instability. The formation of these ~3000 surface lakes has been attributed to the saturation of the ice shelf's firn layer, making it impermeable (Kupiers Munneke et al., 2014; Leeson et al., 2020). Given this possible role of surface water on ice-shelf stability, it is important to monitor changes in the area and volume of surface meltwater systems across ice shelves, and compare any trends with those observed at Larsen B prior to its collapse.

Kingslake et al. (2017) identified numerous pervasive surface meltwater systems across many of Antarctica's ice shelves. Meltwater production is often highest around grounding lines, driven by high net shortwave radiation associated with low albedo blue ice areas, high net longwave radiation around nunataks, and high sensible heat transfer from adiabatic warming of katabatic (Lenaerts et al., 2017) and foehn winds (Bell et al., 2018; Datta et al., 2019). Ice-shelf hydrological systems may then take several forms as meltwater may: (i) form surface streams and flow downslope (e.g. Liston and Winther, 2005; Bell et al. 2017); (ii) collect in surface lakes (e.g. Langley et al. 2016); (iii) percolate into the sub-surface and refreeze (Luckman et al., 2014; Hubbard et al., 2016; Bevan et al., 2017); (iv) percolate into the subsurface and flow laterally (Winther et al., 1996; Liston et al., 1999); or (v) percolate into the subsurface and form sub-surface lakes and reservoirs (e.g. Lenaerts et al. 2017). Despite the identification of pervasive meltwater systems, very little is known about their spatial and temporal evolution, both between and within melt seasons (Arthur et al., 2020). Furthermore, while surface water ponding and the formation of lakes have been implicated in past ice-shelf collapse (Scambos et al., 2003; Banwell et al., 2013), the formation of surface water streams that route water quickly to the ice-shelf front may not necessarily cause instability but rather mitigate against potential surface meltwater-driven collapse (Bell et al., 2017; Banwell, 2017). Thus, whether future projected increased surface melt on ice shelves forms lakes or flows rapidly to the ocean via streams has important implications for future ice-shelf stability and potential collapse. To better understand the behaviour of surface meltwater lakes and streams, it is important to investigate their spatial and temporal

evolution across entire ice shelves through entire summer melt seasons and over multiple
melt seasons.
In this paper, our objective is to develop a tool that can identify surface meltwater bodies on
Antarctic ice shelves and track their evolution over time. We build on the work of Pope et al.
(2016) and Selmes et al. (2011, 2013) and especially Williamson et al. (2017; 2018a), who
developed and used the FAST algorithm for tracking lakes on the Greenland Ice Sheet
(GrIS) using MODIS imagery. More specifically, we have adapted the FASTER algorithm of
Williamson, et al. (2018b) and Miles et al. (2017), who adapted the FAST method to track
GrIS lakes from the higher resolution Landsat 8 and Sentinel-1 and -2 imagery.
These previous methods need adapting for application on Antarctic ice shelves for three
main reasons. First, to account for the observed differences in the geometry of surface
meltwater bodies compared to those on the GrIS. Second, to recognise the marked
geometry changes that occur over time on Antarctic ice shelves, including the joining of
water bodies and the enveloping of some water bodies by others. Third, to identify the
apparent transfer of surface melt water over large distances across ice shelves. In
Greenland, the majority of surface water bodies form in surface depressions that result from
undulations in the bedrock topography and ice flow (Echelmeyer et al., 1991; Sergienko,
2013), and therefore these water bodies evolve in the same location on an inter- and intra-
annual basis (Banwell et al., 2014; Bell et al., 2018). By contrast, the location of surface
water bodies on Antarctic ice shelves reflects variations in the surface topography, which
are controlled by a combination of factors including (i) basal channels formed by ocean
melting (Dow et al., 2018), (ii) basal crevassing (McGrath et al., 2012), (iii) the development
of ice flow-stripes in the grounding zone (Glasser and Gudmundsson, 2012), and (iv) suture
zone depressions (Bell et al., 2017). In Antarctica, these factors result in a wide range of
surface water body geometries; from circular forms, to long linear features that can traverse
significant distances across an ice shelf, and might therefore have significant implications
for the lateral transfer of surface meltwater.
Here, we advance the work of Williamson, *et al.* (2018b) and Miles *et al.* (2017) to produce
'FASTISh', a Fully Automated Supraglacial Lake and Stream Tracking Algorithm for Ice
Shelves. We adapt the FASTER algorithm for use with Landsat 8 and Sentinel-2 data to
make it applicable to Antarctic ice shelves. Such adaptations include: (i) assigning
approximate depths to pixels with floating ice cover; (ii) acknowledging the geometric
variability of surface water bodies across Antarctica and the impact this variability has on
the lateral transfer of surface meltwater by categorising water bodies as either circular or
linear; (iii) assigning each water body that is tracked over the melt season to one of four
categories (always circular, always linear, simple transitions (from circular to linear or vice
versa) and envelopment transitions (where water bodies spread and merge with neighboring
circular and linear water bodies to form a new water body, or where a water body splits into
smaller circular and linear water bodies)) to quantify and illustrate the interaction between
individual water bodies as the melt season progresses. We then apply the FASTISh

algorithm to the Nivlisen Ice Shelf, Antarctica, for the 2016-2017 melt season; the first full melt season to have data coverage over the ice shelf from both Landsat 8 and Sentinel-2.

## 2 Study Area

The Nivlisen Ice Shelf (70.3 °S, 11.3°E), is situated in Dronning Maud Land, East Antarctica, between the Vigrid and Lazarev ice shelves (Fig. 1). It has a surface area of 7,600 km$^2$, and is ~ 123 km wide by 92 km long. Ice thickness ranges from 150 m at the calving front to ~ 700 m towards the ice shelf's grounding line in the southeast, and it exhibits flow velocities of around 20 m a$^{-1}$ to 130 m a$^{-1}$ (Horwath et al., 2006). To the southeast of the Nivlisen Ice Shelf, there is a blue ice region maintained by katabatic winds, which extends in a south easterly direction for ~ 100 km (Horwath et al., 2006). This blue ice region is characterised by ablation, and adjoins the exposed bedrock nunatak (called Shirmacheroasen), which is positioned where the ice shelf meets the inland ice (Horwath et al., 2006) (Fig.1). Beyond this blue ice region, towards the north, the ice shelf transitions into an accumulation zone as the firn layer thickens (Horwath et al., 2006). In the 2016-2017 melt season, mean daily near-surface temperatures on the Nivlisen Ice Shelf ranged between ~ -25°C and 2°C, and 1.6 % of the study area was occupied by a surface water body at least once during this time. The Nivlisen Ice Shelf was selected for this study as: i) pervasive surface meltwater features have previously been identified here in optical satellite imagery, showing evidence of widespread melt ponding in both circular and linear water bodies (Kingslake et al., 2017); ii) these meltwater features have shown significant development over a melt season, as source lakes upstream of the grounding line appeared to drain laterally, rapidly flooding large areas of the ice shelf  (Kingslake et al., 2015); and iii) the ice shelf is relatively small, allowing quick and efficient development and application of FASTISh before its use more widely across larger Antarctic ice shelves.

## 3 Methods

There are four main components to the FASTISh algorithm: i) delineating water body areas; ii) calculating water body depths and volumes; iii) categorising water bodies as either circular or linear based on their geometries; iv) tracking individual water bodies and measuring their changing dimensions and geometries over time (Fig.2). These will be discussed in sections 3.2 to 3.5 respectively, once the pre-processing steps applied to the imagery used have been outlined (section 3.1).

**3.1 Images and Pre-Processing**

**3.1.1 Landsat 8**

12 Landsat 8 scenes with minimal cloud cover, from between 1$^{st}$ November 2016 and 24$^{th}$ March 2017, and each partially covering the ice-shelf extent, were identified and downloaded from the USGS Earth Explorer website (https://earthexplorer.usgs.gov) (Fig. S1). Each scene was downloaded as a Tier 2 data product, in the form of raw digital numbers

(DN). Bands 2 (blue), 3 (green), 4 (red) and 8 (panchromatic) were used for this study (Fig. 2). Bands 2, 3, and 4 have a 30 m spatial resolution, and Band 8 has a 15 m spatial resolution. Image scene values were first converted from DN to Top-of-Atmosphere (TOA) reflectance values. Typically, Landsat scenes are converted to TOA reflectance values using a single solar angle over the whole image scene. However, here we correct each pixel for the specific solar illumination angle, based on metadata stored in the .ANG file, and using the 'Solar and View Angle Generation Algorithm' provided by NASA (https://landsat.usgs.gov/sites/default/files/documents/LSDS-1928_L8-OLI-TIRS_Solar-View-Angle-Generation_ADD.pdf). Converting from DN to TOA values on a per-pixel basis is imperative when mosaicking and comparing images obtained at high latitudes, as the solar angle at the time of acquisition can vary significantly across each scene due to the large change in longitude.

For each Landsat scene, a cloud mask was generated and downloaded from Google Earth Engine (GEE) using the 'Simple Cloud Score Algorithm' (ee.Algorithms.Landsat.simpleCloudScore). The simple cloud score algorithm assigns a 'cloud score' to every pixel in the image based on the following criteria: (i) brightness in bands 2 (blue), 3 (green), 4 (red); (ii) brightness in just band 2 (blue); (iii) brightness in bands 5 (near infrared), 6 (shortwave Infrared 1) and 7 (shortwave infrared 2); and iv) temperature in band 10 (thermal). The algorithm also uses the Normalised Difference Snow Index (NDSI) to distinguish between clouds and snow, which prevents snow from being incorrectly incorporated in the cloud mask. The NDSI was developed by Hall et al. (2001) to distinguish between snow/ice and cumulus clouds and is calculated from the following bands:

$$NDSI = (Blue - Near\ Infrared\ 1)/(Blue + Near\ Infrared\ 1) \qquad (1)$$

We found the 'simple cloud score algorithm' to be the most effective cloud masking method for Landsat 8 images, as it assesses each pixel using multiple criteria, making it more effective than any single band threshold. Prior to implementing the FASTISh algorithm, each Landsat scene and corresponding cloud mask was clipped to the study area extent in ArcGIS using the batch clip process. Clipping each scene to the same extent is required when comparing images through the FASTISh algorithm, as tracking individual features over time requires images with a consistent spatial reference frame to determine the location of each water body. The 12 scenes formed six pairs (Fig. S1), with two scenes per day each covering part of the ice shelf. Each scene pair was mosaicked using ArcGIS's 'mosaic to new raster' tool to produce six images providing near-complete coverage of the ice shelf for six days of the 2016-2017 melt season (Fig. S1). All images were projected into the 1984 Stereographic South Pole co-ordinate system (EPSG: 3031).

### 3.1.2 Sentinel-2

20 Sentinel-2A level-1C scenes obtained between 1st November 2016 and 31st March 2017 with minimal could cover were downloaded from the Copernicus Hub web site (https://scihub.copernicus.eu) (Fig. S1). Bands 2 (blue), 3 (green), 4 (red), and 11 (short

wave infrared (SWIR)) were used. Bands 2, 3, and 4 have a spatial resolution of 10 m, and band 11 a spatial resolution of 20 m. The Sentinel-2 data for all bands were downloaded as TOA reflectance values, and were divided by the 'quantification value' of 10,000 (from metadata), to convert the numbers into values that lie within the 0 to 1 range (Traganos et al., 2018). We applied this conversion to bands 2, 3 and 4 as these are the bands used to identify water and calculate its depth, and their values need to be comparable to the values provided by Landsat 8. Each downloaded scene was clipped, mosaicked to produce images with full coverage of the ice shelf, and then re-projected to the WGS 1984 Stereographic South Pole co-ordinate system (EPSG: 3031), in line with the Landsat scenes. As the simple cloud score algorithm had not been adapted for application to Sentinel-2 imagery at the time of writing, we computed a cloud mask for each image using a thresholding approach, whereby pixels were categorised as cloudy if the SWIR band value was > 10,000. This threshold was selected through visually assessing the effectiveness of various thresholds against the corresponding RGB scenes. As the resolution of the original SWIR band was 20 m, the resultant cloud masks were resampled using nearest neighbor interpolation to 10 m spatial resolution. On two image dates (14[th] November 2016 and 25[th] February 2017), this cloud masking approach was not entirely successful as not all clouds were fully masked. Additional individual masks were manually digitised in ArcGIS to ensure all clouds were masked for these images.

**3.2 Delineating Water Body Areas**

Water body areas were determined using the Normalised Difference Water Index for ice (NDWI$_{ice}$), which has been widely used previously to calculate the distribution of surface meltwater features on the GrIS and on Antarctic ice shelves (e.g. Yang and Smith 2013; Moussavi et al., 2016; Koziol et al. 2017; Macdonald et al. 2018; Williamson et al. 2018b; Banwell et al. 2019). It is calculated from the normalised ratio of the blue and red bands as:

$$NDWI_{ice} = (Blue - Red) / (Blue + Red) \tag{2}$$

These bands were used because water has high reflectance values in the blue band, and there is a relatively large contrast between ice and water in the red band (Yang and Smith, 2013). Studies typically apply a single NDWI$_{ice}$ threshold to an image in order to classify pixels as either 'wet' or 'dry' (e.g. Fitzpatrick et al., 2014; Moussavi et al. 2016; Miles et al. 2017). Across both Greenland and Antarctica, most studies have used a relatively high NDWI$_{ice}$ threshold of 0.25 to map 'deep' water bodies on ice (Yang and Smith 2013, Bell et al. 2017, Williamson et al. 2018b). The same approach was applied to the Nivlisen Ice Shelf in this study in order to facilitate the detection of deep water bodies only. This is important because if too much shallow water and slush is detected, identifying and subsequently tracking individual water bodies over time becomes difficult. Having applied a 0.25 NDWI$_{ice}$ threshold to each image, the resulting water masks were filtered using a two-dimensional 8-connected threshold (i.e. grouping pixels if they were connected by their edges or corners) to identify each individual water body. Water bodies consisting of $\leq$ 2 pixels (Landsat 8) and $\leq$ 18 pixels (Sentinel-2), were removed to ensure only water bodies with an area $\geq$ 1,800 m$^2$

were assessed further. To ensure that pixels with floating ice cover were still included in the analysis, we then used the 'imfill' function within MATLAB to classify any 'dry' pixels situated within a water body as water.

**3.3 Water Body Depth Calculations**

Having identified the extent of water bodies, we use a physically-based approach (Sneed and Hamilton, 2007; Arnold et al., 2014; Banwell et al., 2014, 2019; Pope, 2016; Pope et al., 2016; Williamson et al., 2017, 2018b) based on the original work of Philpot (1989), to calculate pixel water depths. Water depth, $z$, is calculated from:

$$z = \frac{[\ln(A_d - R_\infty) - \ln(R_{pix} - R_\infty)]}{g} \tag{3}$$

where $R_{pix}$ is the satellite-measured pixel reflectance, $A_d$ is the lake-bottom albedo, $R_\infty$ is the reflectance value for optically deep (> 40 m) water, and $g$ is the coefficient associated with the losses made during downward and upwards travel in a water column.

For the Landsat 8 images, pixel water depths were calculated using TOA reflectance data for both the red and panchromatic bands separately, and then averaging these values to give a single final value (Pope *et al.*, 2016; Williamson *et al.*, 2018b). Pope et al. (2016) show that this approach gives the smallest mean difference (0.0 +/- 1.6m) between spectrally-derived and DEM-derived lake depths. However, it should be noted that owing to the rapid attenuation of red light by a water column, this algorithm is only able to retrieve depths up to a maximum of ~ 5 m (Pope et al., 2016). Furthermore, this method assumes: (i) no wind and waves at the water body surface (ii) little to no dissolved/suspended material within the water body, (iii) no inelastic scattering, and (iv) the water body substrate is parallel to the surface and homogenous (Sneed and Hamilton, 2011).

For Landsat 8 images, the panchromatic band was first resampled using bilinear interpolation from 15 m to 30 m spatial resolution to match the resolution of the red band. For the Sentinel-2 images, water body depths were calculated using the TOA reflectance values in the red band only, as there is no equivalent panchromatic band (Williamson *et al.* 2018b). To calculate $A_d$, the mean reflectance value of the second (Landsat) and sixth (Sentinel) rings of pixels outside of each water body was calculated, following a similar approach used by Arnold et al. (2014) and Banwell et al. (2014). The second or sixth ring of pixels surrounding each lake was used to avoid calculating $A_d$ from slushy areas that border each water body; sixth-pixel rings were used for Sentinel-2 images as these represent the same distance away from the water body as two-pixel rings in Landsat images. In very rare cases, wet pixels within a water body could have a reflectance higher than the calculated $A_d$ value, leading to negative water depths. All such pixels were removed from the area and depth matrix (Fig. 2).

Values for $R_\infty$ were assessed on an image-by-image basis by taking the minimum
reflectance value found over optically deep water (the ocean). For images that did not
contain optically deep water, the $R_\infty$ value was set to 0 (Banwell et al., 2019). For Landsat 8
imagery we used a *g* value of 0.7507 for the red band, and 0.3817 for the panchromatic
band (Pope *et al.*, 2016), and for Sentinel-2 imagery, we used a value of 0.8304 (Williamson
*et al.*, 2018b). Pixels in the lake masks that were filled (normally those with a floating ice
cover, see section 3.2) were assigned the mean water depth of their respective water
bodies. Individual water body volumes were calculated by multiplying each pixel area by its
calculated water depth, and then summing across the water body. To facilitate comparisons
between Landsat 8 and Sentinel-2 data, area and depth arrays generated from Landsat 8
images were then resampled to 10 m spatial resolution using nearest neighbour
interpolation.
**3.4 Classifying Water Body Types**
Having produced area and depth masks for each date, each identified water body was
categorised as either circular or linear based on its solidity (defined as the proportion of
pixels of the water body that fall within its convex hull), which was calculated using the
'regionprops' function in MATLAB (Banwell et al. 2014). Linear water bodies have a solidity
score closer to 0 reflecting the smaller proportion of wet pixels within the convex hull due to
likely greater concavity of the edges, whereas more circular water bodies have a solidity
score closer to 1 due to the larger proportion of wet pixels within the convex hull due to the
more convex shape. Here, water bodies with a solidity score $\geq$ 0.45 were classified as
circular, and water bodies with a solidity score < 0.45 were classified as linear. This threshold
was selected by visually assessing the masks generated from thresholds ranging between
0.42 and 0.49, in increments of 0.01, and selecting the threshold that appears to best
distinguish between more circular and more linear water bodies (Fig. 3).
**3.5 Tracking Water Bodies**
A 3D matrix of all water bodies was compiled, recording the area and volume of each water
body over time, as well as whether the water body had a circular or linear geometry (as
defined in section 3.4). To track changes in the area and volume of surface meltwater bodies
throughout the 2016-2017 melt season, a maximum extent mask (Fig. 4b) was also
generated by superimposing the areas of all water bodies identified in each image
(Williamson *et al.*, 2018b). The maximum extent mask was then used to guide the tracking
process. Each individual water body within the maximum extent was prescribed an ID, and
changes to the area and volume of each individual water body over time were tracked within
its maximum extent (Williamson *et al.*, 2018b).
In addition to tracking changes in the area and volume of each water body, the FASTISh
algorithm also tracks the water body type. From this tracking process, four categories were
defined: (i) always circular, (ii) always linear, (iii) 'simple transitions' where a water body is
defined as *either* circular or linear and switches between the two categories (either once or

more than once, and in either direction), and (iv) 'envelopment transitions' where water bodies spread and merge with neighbouring circular and linear water bodies to form new, larger bodies, or where larger bodies split into smaller circular and linear water bodies. This final category allows us to track the development of large surface water bodies across the ice-shelf surface as it identifies smaller water bodies being subsumed by larger water bodies as the melt season progresses.

**3.6 Digital Elevation Model**

To aid interpretations of the tracking results produced by the FASTISh algorithm, we used surface elevation data from the Reference Elevation Model of Antarctica (REMA) database (Howat et al., 2019). Figure 4a shows the REMA Digital Elevation Model (DEM) of the ice shelf at 8 m resolution, produced by mosaicking four, 8 m resolution REMA tiles. In addition, a single 2 m REMA data strip from 31$^{st}$ January 2016 was used to extract the elevation profiles along two tracked water bodies, the Eastern System and the Western System, which are introduced in section 4.2.2.

**3.7 Regional climate simulation**

In order to understand how climate variability influences the findings, we analysed results from an atmosphere-only regional climate CORDEX (COordinated Regional climate Downscaling Experiment) simulation of Antarctica using the limited-area configuration of Version 11.1 of the UK Met Office Unified Model (MetUM) for the period 2016-2017. The MetUM is a weather prediction model, which uses a semi-Lagrangian semi-implicit scheme for solving the fully-compressible, non-hydrostatic, deep-atmosphere equations of motion (Walters et al., 2017).

The setup of the MetUM is similar to that used by Mottram et al. (2020), with the exception that the horizontal resolution for the limited-area Antarctic domain has been increased from 50 to 12 km (and consists of 392 × 504 grid points). The Antarctic domain uses the regional atmosphere mid-latitude (RA1M) science configuration (Bush et al., 2019), a rotated latitude-longitude grid in order to ensure that the grid points are evenly spaced, and 70 vertical levels up to an altitude of 40 km.

The required start data and lateral boundary conditions for the Antarctic domain are supplied by a global version run of the MetUM at N320 resolution (640 × 480 grid points, equivalent to a horizontal resolution of 40 km at mid-latitudes), which is itself initialised by ERA-Interim atmospheric reanalysis (Dee et al., 2011). The model is used to provide a series of 12 to 24 hr forecasts, provided every 12 hrs, for the period 20151231T1200Z to 20171230T0000Z, i.e. the initial 12 hrs of each forecast is discarded as spin-up. We extracted a continuous forecast time-series for the period November 2016 to April 2017. We extracted daily mean and daily maximum near-surface diurnal air temperatures (at a height of 1.5 m above the ground) for the model grid-point immediately to the north of Schirmacheroasen.

## 4 Results

**4.1 Spatial Extent and Distribution of Surface Water Bodies**

The seasonal evolution of meltwater bodies during the 2016-2017 summer is shown in Figure 5. The surface meltwater system transitions from a series of small isolated water bodies clustered towards the grounding line (Fig 5A), to a connected system dominated by two linear water bodies with a length of (a) ~ 20.5 km and (b) ~ 16.9 km that propagate towards the ice-shelf front (Fig 5D).

For example, on 11th December 2016, few meltwater bodies exist, and they are predominantly clustered within the blue ice region towards the grounding line in the south-west (Fig. 5A). The majority of these water bodies exist as distinct entities, and do not connect to one another. Some meltwater ponds are identified in close proximity to the nunatak. The total volume and area of all surface meltwater bodies on the 11th December is $2.8 \times 10^6$ m$^3$ and $2.8 \times 10^6$ m$^2$ respectively (Table 1). The mean water depth is 1.0 m, and the maximum water depth is 3.4 m (Table 1). By 17th December (Fig. 5B), there has been a marked increase in the total volume ($3.2 \times 10^7$ m$^3$) and area ($4.7 \times 10^7$ m$^2$) of surface meltwater, held in both circular and linear surface water bodies (Table 1). The mean water depth is 0.7 m and the maximum water depth is 3.1 m (Table 1). Several of the previously isolated ponds have coalesced in some of the main topographic lows. The spatial extent of the surface water bodies extends ~ 2 km further towards the ice-shelf front. In addition, some water bodies have begun to develop towards the eastern edge of the grounding line in a blue ice region.

A marked shift in the surface meltwater system is identified by 27th December (Fig 5C), as two large linear water bodies have formed along the north-south axis (labelled a and b in Fig. 5C). The Western linear water body (a) is ~ 6.5 km long and ~ 10 km from the Eastern linear water body (b), which is ~ 8.5 km long and proximal to the surface lakes on the ice shelf's eastern margin (Fig. 5C). Overall, there are fewer isolated lakes towards the grounding line, and the majority of the surface meltwater is proximal to the two large linear systems, at elevations of ~ 60 m to 65 m (Fig. 4). The total volume and area of all surface meltwater bodies is $4.9 \times 10^7$ m$^3$ and $5.4 \times 10^7$ m$^2$ respectively (Table 1). The mean water depth of all identified water bodies is 0.9 m and the maximum water depth is 4.7 m (Table 1).

By 26th January 2017 (Fig. 5D), the total volume and area of surface meltwater reaches a peak for the summer, at $5.5 \times 10^7$ m$^3$ and $9.1 \times 10^7$ m$^2$ respectively (Table 1). This is facilitated by the enlargement of the two large linear systems, which involves the flooding of topographic lows as water appears towards the firn further north on the ice shelf. These linear systems are now (a) ~ 20.5 km and (b) ~ 16.9 km in length, and have a mean depth of (a) 0.8 m and (b) 0.7 m. The mean depth of all water on 26th January 2017 is 0.6 m and the maximum water depth is 3.3 m (Table 1).

By 13th February (Fig 5E), the two large linear systems remain prominent on the ice shelf,
but they have lost area, depth and volume at their southern ends. The mean water depth of
all water is 0.6 m and the maximum water depth is 4.3 m (Table 1). The total volume and
area of surface meltwater bodies falls to 3.7 x $10^7$ m$^3$ and 6.3 x $10^7$ m$^2$ (Table 1), reflecting
the shrinkage of the two linear systems.
**4.2 Tracking Results**
Of the 1598 water bodies identified and tracked within the maximum extent matrix, 1458
(91%) are defined as always circular, 42 (3%) are identified as always linear, 51 (3%) are
defined as simple transitions, and 47 (3%) are categorised as envelopment transitions.
Water bodies that are always circular are predominantly clustered further south on the ice
shelf towards the grounding line, while water bodies defined as envelopment transitions are
found further north, towards the ice-shelf front (Fig. 6).
*4.2.1 Total Area and Volume of Tracked Surface Water Bodies*
For each of the tracked water body categories, Table 2 shows the maximum area and
volume, and the corresponding dates on which these maxima were reached. The minimum
area and volume for all tracked categories is zero on 14th November 2016, as no deep
surface melt water was detected on that date. Although 91% of water bodies identified are
classified as circular, they do not dominate the total area or volume of surface meltwater
(Fig. 7). Conversely, the envelopment transitions, of which there are only 47 in total, peak
at 8.0 x $10^7$ m$^2$ in area and 4.5 x $10^7$ m$^3$ in volume on 26th January 2017, over a month later
than the peaks in area and volume recorded for the other three categories. These
envelopment transitions dominate the total area and volume signals for 'all water bodies',
which also reach their maxima on 26th January (Table 2, Fig. 7). Between 17th December
2016 and 27th December 2016 'all water bodies' are effectively deepening, as their mean
depth increases whilst the total area increases, whereas between the 27th December and
the 26th January 'all water bodies' are effectively spreading, as their mean depth decreases
whilst total area increases. (Table 1, Fig. 7).
*4.2.2 Tracking Individual Water Bodies*
In addition to quantifying total surface water area and volume for each of the four water body
categories (Fig. 7), the FASTISh algorithm also tracks changes in the area and volume of
*individual* water bodies. Over the 2016-2017 melt season, the two largest envelopment
transitions, referred to as the Western System (WS) and the Eastern System (ES) hereafter,
propagate towards the ice-shelf front as the melt season progresses, and contain 62.6 % of
the total surface water volume on 26th January 2017. The remainder of this sub-section
focuses solely on presenting the tracking results for these two water bodies.

The WS is active between 11[th] December 2016 to 25[th] February 2017. The area and volume
of meltwater within the WS reaches a maximum of 4.6 x 10$^7$ m$^2$ and 2.5 x 10$^7$ m$^3$ respectively
on 26[th] January 2017 (Fig. 8). The ES has a shorter lifespan, and is active between 27[th]
December 2016 and 25[th] February 2017 (Fig. 8). The area and volume of the ES peaks at
1.9 x 10$^7$ m$^2$  and 9.6 x 10$^6$ m$^3$ on the 26[th] January 2017.  Figure 9 shows the surface
elevation profiles for the WS and the ES, which are extracted from the maximum extent
mask (see section 3.5). Both systems are characterised by a surface sloping downwards
towards the ice-shelf front. The WS has a very shallow slope, with the elevation decreasing
by ~ 2 m over the 25.7 km profile (Fig. 9a); the ES is slightly steeper, showing a ~ 7 m
decrease in elevation over its 27 km profile (Fig. 9b).
*4.2.3 Identifying Individual Lake Freeze Through/Drainage Events.*
Previous studies have attempted to identify rapid drainage events, defined as events where
lakes lose > 80 % of their maximum volume in ≤ four days (e.g. Fitzpatrick et al., 2014; Miles
et al., 2017; Williamson et al., 2018b). Here, however, the temporal resolution of available
imagery for the Nivlisen Ice Shelf is not high enough to allow this. Therefore, we used the
calculated volume time series to identify water bodies in the 'always circular' category that
lost > 80 % of their maximum volume over the full melt season, through either drainage or
freeze through.  We focus solely on the 'always circular' category to better understand the
local loss of surface melt water in seemingly isolated and stationary water bodies. These
events are referred to as 'loss events' hereafter.
Figure 10 shows the loss in water volume through freeze-through or drainage for the 'always
circular' category over the melt season, together with the seven day moving average for the
mean daily and daily maximum near-surface air temperatures over the ice shelf from the
MetUM simulation. This shows that 805 lakes have a 'loss event' by 18[th] December 2017,
losing a total volume of 1.5 x 10$^7$ m$^3$, which occurs following sustained relatively warmer
atmospheric conditions since the beginning of December 2016, e.g. characterised by daily
maximum near-surface air temperatures reaching 0$^o$C.
**5 Discussion**
**5.1 Spatial and Temporal Distribution of Surface Meltwater Bodies**
In the early melt season, surface meltwater on the Nivlisen Ice Shelf ponds in small surface
lakes that form in relatively flat areas towards the grounding line, in close proximity to
Shirmacheroasen and the blue ice regions (Figs. 4 and 5). This initial generation of surface
meltwater is likely driven by regional wind patterns and the effects of local ice-albedo, as the
relatively low albedo of the blue ice can lead to increased local melt rates (Lenaerts et al.,
2017; Bell et al., 2018; Stokes et al., 2019). Furthermore, areas of lower elevation towards
the grounding line are likely to be exposed to katabatic winds, which can result in near-
surface temperatures that are 3 K greater than temperatures further up-ice and down-ice
(Lenaerts et al. 2017). These persistent katabatic winds can also result in the production of

blue ice regions, as snow is eroded from the ice-shelf surface (Lenaerts et al., 2017). Our results for the early melt season on the Nivlisen Ice Shelf therefore support the findings of Kingslake et al. (2017) who found, for a variety of ice shelves around Antarctica, that 50 % of the ice-shelf drainage systems are either within 8 km of rock exposures, or within 3.6 km of blue ice surfaces.

Seasonal variations in the amount of surface meltwater on the Nivlisen Ice Shelf are driven by temperature fluctuations, with increases in surface water area and volume corresponding with rising mean daily near-surface temperatures and daily maximum near-surface temperatures (Fig. 10). However, as the melt season progresses, there is a transition to a connected surface drainage network, which facilitates a progressive transfer of surface meltwater away from the grounding line towards the ice-shelf front. As mean daily and daily maximum temperatures rise and surface water bodies increase in area and volume (Fig.10), they grow, merge with nearby water bodies, and form new extended networks of surface water on the ice-shelf surface. While rising near-surface temperatures are a strong control on the amount of surface meltwater, the direction and extent of the identified lateral water transfer is controlled by the ice shelf's surface topography (Fig. 4b). Over the course of the melt season, the area and volume of surface meltwater decreases in the regions close to the grounding line, and increases in more distal parts of the ice shelf.

The development of the two largest enveloping water bodies (WS and ES) dominate the transition to a generally more connected drainage network. This is because these systems facilitate large-scale transfer of water across the shelf, as water ponds within linear depressions. The ES and WS appear to be fed by smaller circular and linear surface meltwater bodies, and as the area and volume of the ES and WS increases, they spread and envelope nearby water bodies. Smaller water bodies likely contribute surface melt to the ES and WS by (i) overtopping their local basin sides and flowing over impermeable ice, which may be refrozen surface or shallow subsurface meltwater from previous years (Kingslake et al., 2015) or (ii) percolating into the firn pack and spreading laterally towards the ES and WS. However, the 'pulse' forward of the ES and WS between 27[th] December 2016 and 26[th] January 2017 does not appear to be due to a breach of a topographic 'lip' or 'dam' (Fig. 9). It is therefore likely to primarily be the result of increased meltwater production, resulting in saturation of the surrounding firn pack, which may bring it up to isothermal conditions, thereby facilitating further melt and lateral transfer.

By 26[th] January 2017, the ES and WS are the dominant features within the entire Nivlisen Ice Shelf meltwater system, together holding 62.6% of the surface meltwater volume. On this date, the ES and WS reach a length of ~ 16.9 km and ~ 20.5 km respectively, although unlike observations on the Nansen Ice Shelf (Bell et al., 2017), they do not facilitate the export of surface meltwater off the ice-shelf front via a waterfall. Instead, both systems always terminate at least ~ 35 – 55 km from the ice-shelf front, suggesting that the water percolates into the surrounding firn in that area of the ice shelf. This interpretation is supported by Figure 11 which shows a Sentinel-1 SAR image (Fig 11b), from 26th January 2017 together with the Sentinel-2 image (Fig 11a). Areas of low backscatter

(appearing as dark areas in Figure 11b) extend across the grounding line onto the upper
part of the ice shelf. Whilst areas of low backscatter may result from relatively small dry-
snow grain sizes, shallow dry-snow depths to underlying rougher surfaces, high surface
roughnesses, or complex internal stratigraphies (Rott and Mätzler, 1987; Sun et al., 2015),
it seems more likely that areas of low backscatter north of the blue ice areas represent
saturated firn and/or surface melt (Bindschadler and Vornberger, 1992; Miles et al., 2017).
Areas of low backscatter clearly extend beyond areas of visible surface melt in the optical
imagery, indicating the presence of subsurface meltwater. For example, there are
prominent areas of low backscatter (~ - 5 to -15 dB) extending ~ 10 km north of both the
ES and WS as detected by FASTISh (Fig 11b). This shows that the linear water features
visible in the optical imagery are part of much larger water bodies, with a lot of the water
existing as slush at the surface or in the shallow subsurface.
Whilst the drainage system currently observed on the Nivlisen Ice Shelf does not transfer
surface meltwater all the way to the ice-shelf front, it is plausible that such a system could
develop in the future as the quantity of surface meltwater produced increases. Whilst the
water may pond, (possibly resulting in eventual hydrofracture and ice-shelf collapse), the
ES and WS may also evolve quickly and efficiently, over increasingly saturated firn layers,
to allow water to flow off the ice-shelf front, thereby exporting some excess meltwater and
mitigating the potential threat to the ice shelf (Bell et al., 2017; Banwell, 2017).
Overall, 1.6 % of the Nivlisen Ice Shelf is occupied by some form of surface meltwater body
at some point during the 2016-2017 melt season, and over those areas, the mean water
depth is 0.85 m. Comparatively, prior to its collapse, 5.3 % the Larsen B Ice Shelf was
covered by a surface meltwater body, and the mean water depth was 0.82 m (Banwell et
al., 2014). Whilst the mean water body depths between the Larsen B and Nivlisen Ice
Shelves are comparable, the spatial distributions of these water bodies, and the proportion
of the ice shelf that they cover, are different. Surface water bodies were distributed relatively
evenly across the entire surface of Larsen B before it collapsed, whereas surface water
bodies are predominantly clustered towards the grounding line on the Nivlisen Ice Shelf, and
the transfer of surface melt towards the ice-shelf front and across snow/ firn-covered regions
is predominantly facilitated by the larger WS and ES. The development of these large, linear
water bodies is likely facilitated by ice-shelf surface topography, and allows the transfer of
summer meltwater towards the ice-shelf front. This large scale lateral transfer of meltwater
is likely further facilitated as the ES and WS develop over frozen meltwater paths from
previous years (Kingslake et al., 2015).
**5.2 Loss of Water Volume from Circular Surface Water Bodies**
The loss of $1.5 \times 10^7$ m$^3$ of surface water from the circular water bodies by 27[th] December
2017 follows sustained relatively warmer atmospheric conditions since the beginning of
December 2017 (Fig. 10), and coincides with an increase in the total surface water volume
on the ice shelf (Fig 10b). In particular, we see an increase in the volume of water held within
the enveloping water bodies, which continues to increase up to a maximum of $4.5 \times 10^7$ m$^3$

on 26[th] January 2017 (Fig. 7). It is likely, therefore, that the loss of water from circular water bodies at this early stage in the melt season signifies the lateral transfer of water away from the small 'isolated' bodies near the grounding line into the large enveloping water bodies which hold and transport the surface meltwater across the ice shelf to more distal regions. This lateral transfer of water may be occurring through two mechanisms: (i) the over-topping of surface lakes, which results in the formation of shallow channels that connect water bodies and facilitate the transfer of water towards the ice-shelf front (e.g. Banwell et al., 2019), or (ii) the gradual percolation of surface meltwater into the cold firn pack, which reduces the firn air content (FAC) of a region (Lenaerts et al., 2017), therefore creating an impermeable surface over which water can flow (e.g. Kingslake et al., 2015). The firn may also become saturated enough to be isothermal, therefore melting and facilitating the flow of upstream ponded meltwater. This is particularly likely to occur near surface depressions such as those that are later occupied by the WS and ES.

**5.3 Potential Implications for Ice-Shelf Stability**

It is expected that the area of coverage and volume of surface meltwater on Antarctic ice shelves will increase into the future, in line with rising atmospheric temperatures (Bell et al., 2018; IPCC, 2019; Kingslake et al., 2017; Siegert et al., 2019). This surface water may have significant implications for ice-shelf stability, as meltwater accumulation can lead to hydrofracture which could subsequently result in the collapse of an ice shelf, as seen on the Larsen B Ice Shelf in 2002 (Robel and Banwell, 2019; Banwell et al., 2013). An ice shelf may become increasingly vulnerable to hydrofracture if its FAC is reduced (Lenaerts et al., 2017). On ice shelves like Nivlisen, where large-scale lateral water transfer prevails, meltwater is delivered to locations that may otherwise not receive or experience much melt (Bell et al., 2017), and the FAC of these locations will, in turn, be reduced, increasing their susceptibility to surface meltwater ponding and hydrofracture.

Surface meltwater re-freezing at the end of the melt season will also act as a significant source of heat, and the lateral transfer of surface melt could cause increased warming of the ice shelf and possible weakening in areas which currently do not experience significant summer melt. Were the maximum volume of surface meltwater we observe on the Nivlisen Ice Shelf in the 2016-2017 melt season ($5.5 \times 10^7$ m$^3$) to re-freeze over the maximum area of surface meltwater ($9.1 \times 10^7$ m$^2$), it would release an amount of energy equivalent to 49 days of potential solar energy receipts (calculated using the methods of Arnold and Rees (2009)), assuming an ice surface albedo of 0.86; the mean value calculated for a water-free distal area of the ice shelf. Furthermore, large-scale lateral water transfer and subsequent ponding may lead to ice-shelf flexure (and therefore potential fracture) at locations that may have otherwise not been affected by flexure in response to meltwater loading (Banwell et al., 2013, 2019; Macayeal and Sergienko, 2013). However, evidence of lateral water transfer and export off the Nansen Ice Shelf has highlighted the potential for surface drainage systems to mitigate some of these meltwater-driven instabilities (Bell et al., 2017).

## 6 Conclusions

We have adapted the pre-existing FASTER algorithm, developed for studying lakes on the GrIS (Williamson et al., 2018b), so that we can identify and track the area, depth and volume of water bodies across Antarctic ice shelves. We refer to this new algorithm as FASTISh, and have used it to study the changing geometry and spatial patterns of water bodies across the Nivlisen Ice Shelf in the 2016-2017 melt season. In total, we identify and track 1598 water bodies on the ice shelf over the course of the melt season. Surface water is initially generated towards the nunatak and blue ice region, in proximity to the grounding line. This region is relatively flat and has a low albedo, and we therefore observe localised ponding of surface meltwater. As the melt season progresses and mean daily and daily maximum temperatures increase, we see a transition from isolated, localised ponding towards the grounding line to a more connected drainage system that is influenced by the ice-shelf topography. The middle of the melt season (e.g. 27th December 2016) is characterised by the progression of surface melt water bodies towards the ice-shelf front, as two large extensive drainage systems (the East System (ES) and West System (WS)) develop in long linear surface depressions. Around the peak of the melt season (26th January 2017), the ES and WS have developed to their largest observed extent, and facilitate the lateral transfer of surface melt up to 16.9 and 20.5 km north, into the firn pack and towards the ice-shelf front. The transfer of surface meltwater to regions on the ice shelf that otherwise experience little surface melt may have implications for the structure and stability of the ice shelf in the future. Our findings could be useful in comparing to IceSat 2 derived lake depths (Fair et al., 2020), in addition to constraining ice-shelf surface hydrology models (Buzzard et al., 2018).

**Code and Data Availability**

The satellite imagery, REMA data, and meteorological data are all open access (see section 3). The MATLAB scripts used to process the data will be freely available from Apollo Repository (https://www.repository.cam.ac.uk/) upon publication.

**Author Contributions**

RLD developed the methodology and scripts, building on the prior work of AGW. NSA developed the script to convert Landsat DN values to per-pixel TOA values. AO performed the Regional Climate Model run using the Met Office Unified Model to provide the meteorological data. RLD conducted all other analysis and wrote the draft manuscript, under the supervision of all other authors. All authors discussed the results and were involved in editing of the manuscript.

**Competing Interests**

The authors declare no competing interests

**Acknowledgements**

We sincerely thank Mahsa Moussavi and Allen Pope for their guidance and many productive discussions over the past two years. Rebecca Dell acknowledges support from a Natural Environment Research Council Doctoral Training Partnership Studentship (Grant number: NE/L002507/1; CASE Studentship with the British Antarctic Survey). Ian Willis acknowledges support from NERC Grant G102130. This paper was written while Ian Willis was in receipt of a Cooperative Institute for Research in Environmental Science (CIRES) Visiting Sabbatical Fellowship and he thanks in particular Waleed Abdalati, Ted Scambos, Kristy Tiampo and Mike Willis for their hospitality. Alison Banwell acknowledges support from a CIRES Visiting Postdoctoral Fellowship and a grant from the US National Science Foundation (#1841607) awarded to the University of Colorado, Boulder. AO thanks Tony Phillips (BAS) for converting the MetUM output to daily averaged fields. DEMs provided by the Byrd Polar and Climate Research Center and the Polar Geospatial Center under NSF-OPP awards 1543501, 1810976, 1542736, 1559691, 1043681, 1541332, 0753663, 1548562, 1238993 and NASA award NNX10AN61G. Computer time provided through a Blue Waters Innovation Initiative. DEMs produced using data from DigitalGlobe, Inc.

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

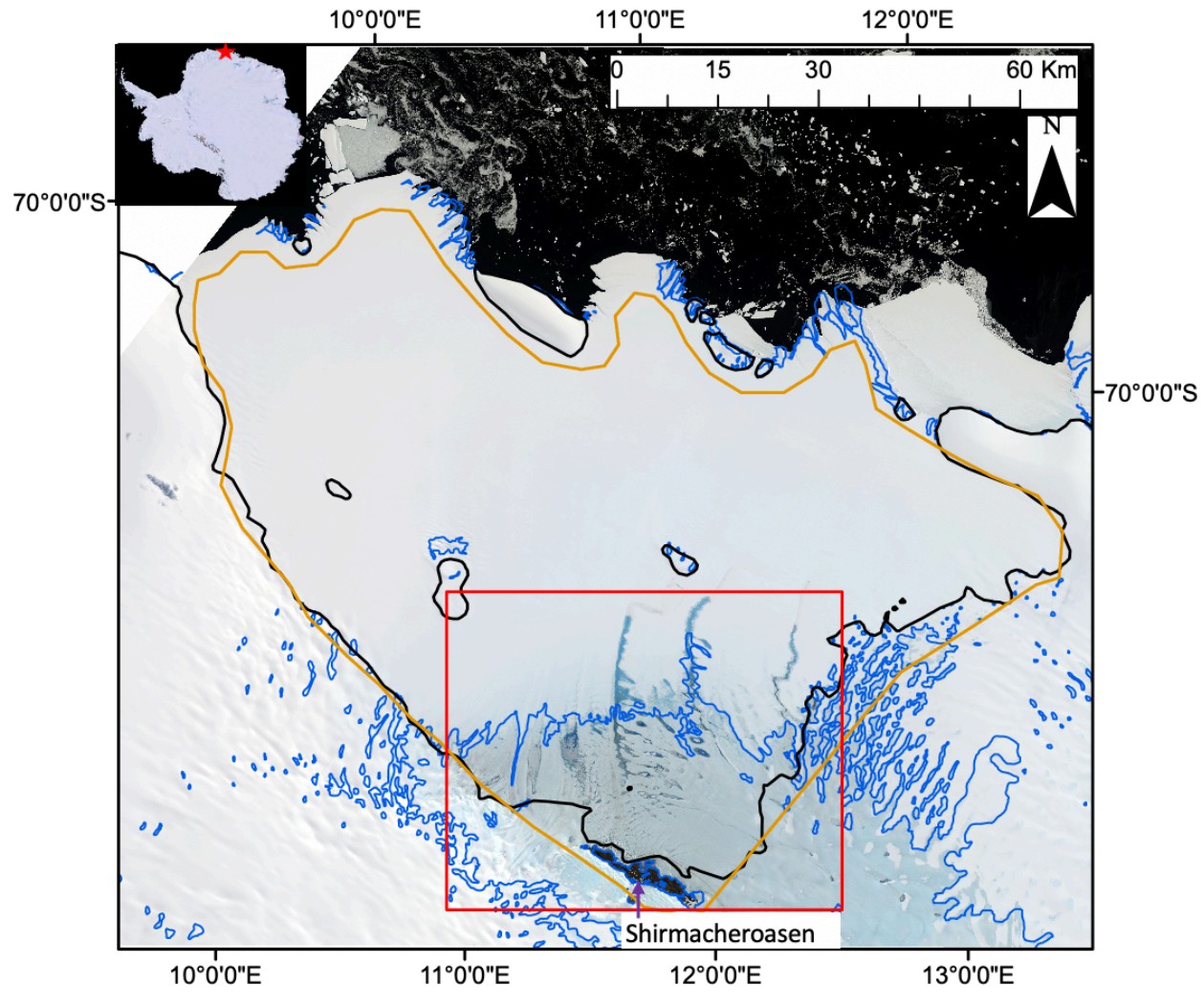

*Figure 1: A map of the study area. The base image is a mosaicked RGB Sentinel-2 image of the*
*Nivlisen Ice Shelf acquired on 26th January 2017. The solid black line marks the grounding line,*
*according to the NASA Making Earth System Data Records for Use in Research Environments*
*(MEaSUREs) Antarctic boundaries dataset* (Mouginot et al., 2017). *The solid blue line represents*
*the blue ice areas in the region according to* Hui et al. (2014). *The solid orange line roughly delineates*
*the ice shelf and shows the study area extent used for this study, and the solid red line marks the*
*area shown in all subsequent figures. The red star on the inset shows the location of the Nivlisen Ice*
*Shelf in the context of an image of Antarctica, which is a mosaic product based on sources from*
*USGS, NASA, National Science Foundation, and the British Antarctic Survey*
*(https://visibleearth.nasa.gov/view.php?id=78 592).*

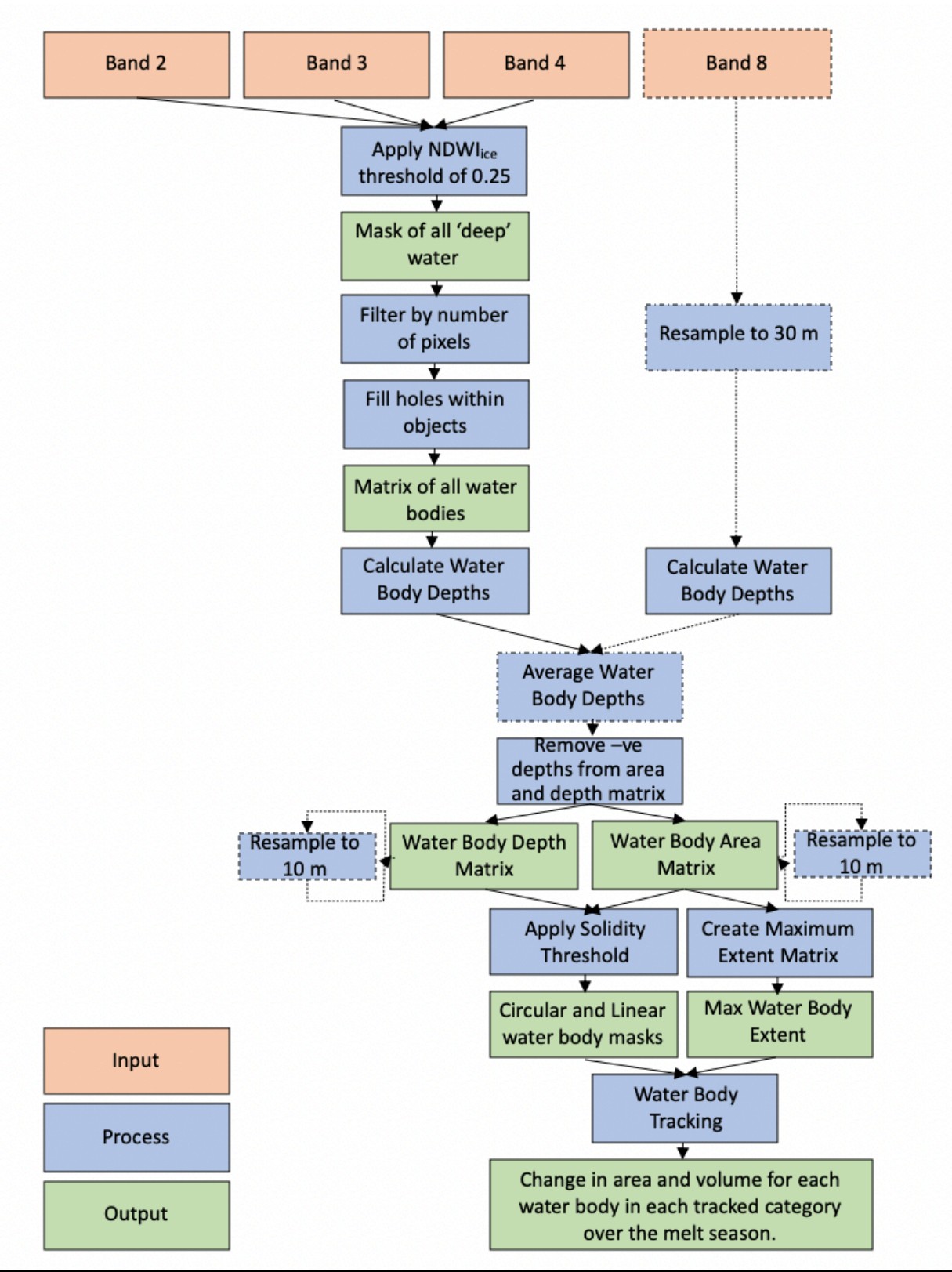



*Figure 2: Workflow detailing the methods applied to both the Landsat 8 and Sentinel-2 images*
*through the FASTISh algorithm in MATLAB. Dashed lines indicate steps that were applied to Landsat*
*8 images only, whereas solid lines indicate steps that were applied to both sets of image types.*
*Modified from Williamson et al. (2018b).*


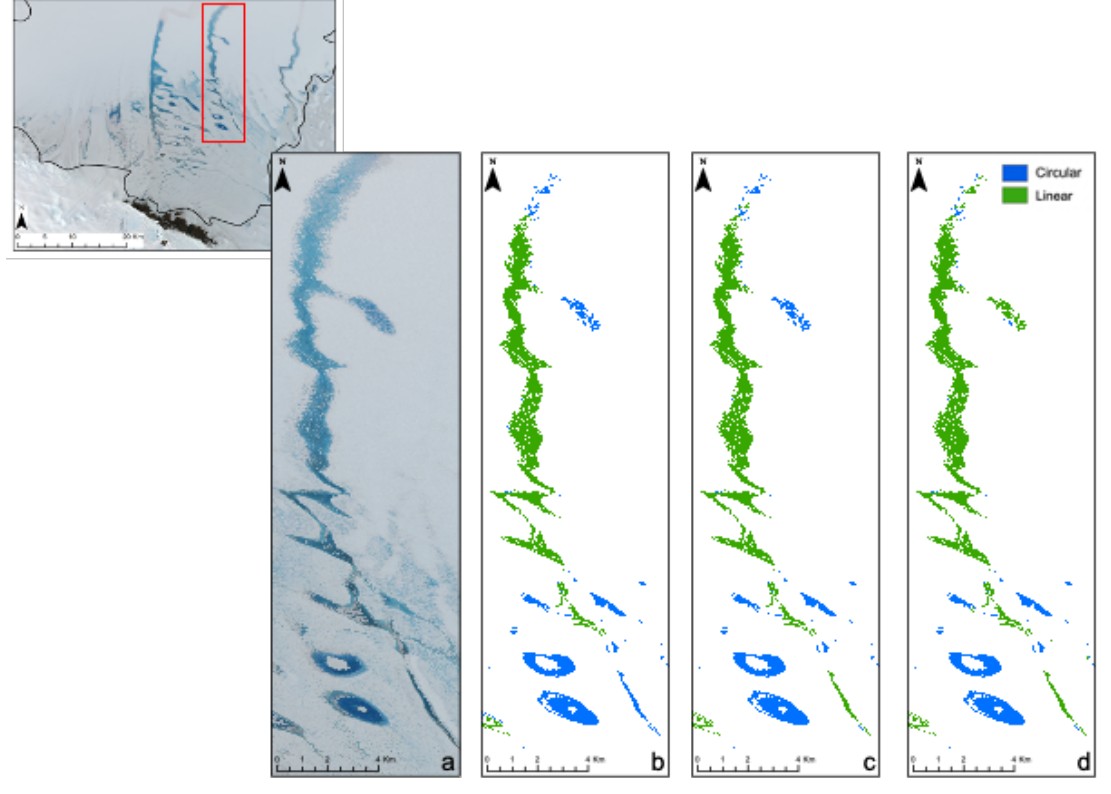



Figure 3: Solidity thresholds applied to water bodies identified on the Nivlisen Ice Shelf. The subset
Sentinel-2 image is from the 26[th] January 2017, and the red box indicates the area shown in
panels a-d. a) shows this area as an RGB, b) shows the water bodies identified and separated into
linear or circular water bodies using a threshold of 0.42, c) a threshold of 0.45, and d) a threshold
of 0.49.


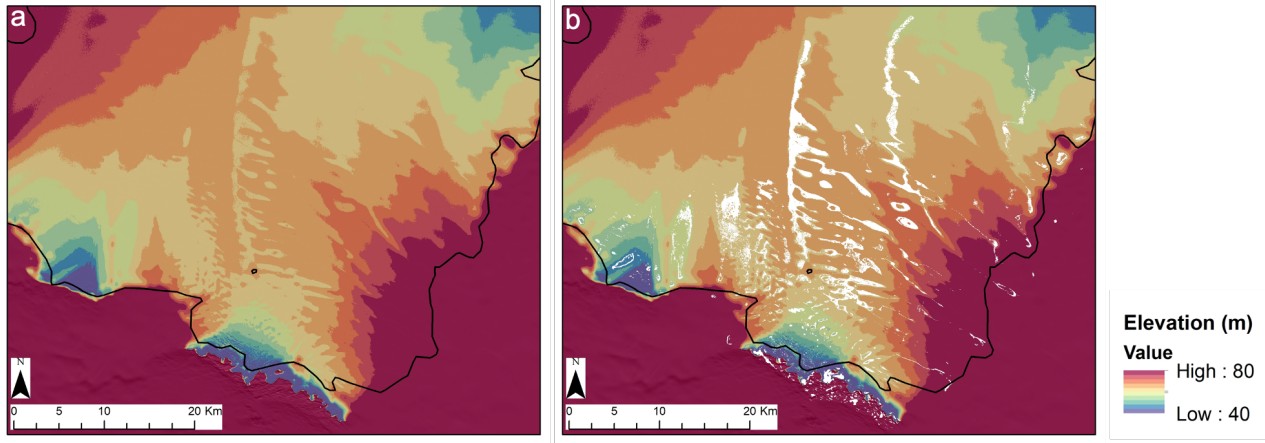



Figure 4: REMA DEM data for the Nivlisen Ice Shelf. a) the DEM; and b) overlain with the maximum
melt extent matrix for the 2016-2017 melt season in white. DEM data sourced from the REMA
dataset (Howat et al., 2019).


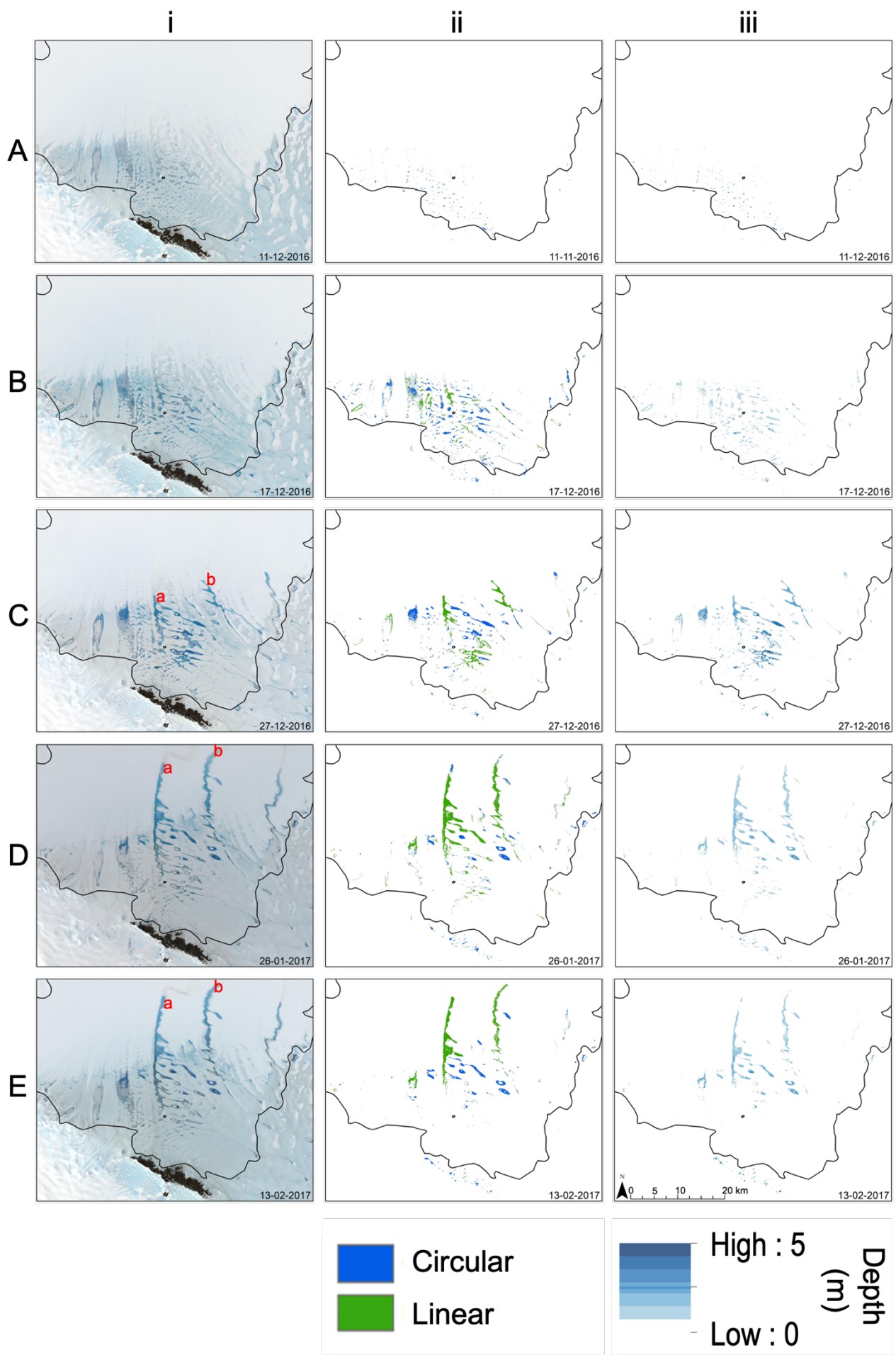


Figure 5: Five of the eleven dates studied in the 2016-2017 melt season (represented by labels A-
E), and their corresponding (i) RGB images, (ii) area masks for circular and linear features, (iii) depth
masks. Date stamps are in the bottom right hand corner of each image. Fig. S2 for all RGB images,
Fig. S3 for all lake and stream area masks and Fig. S4 for all depth masks produced in this study.

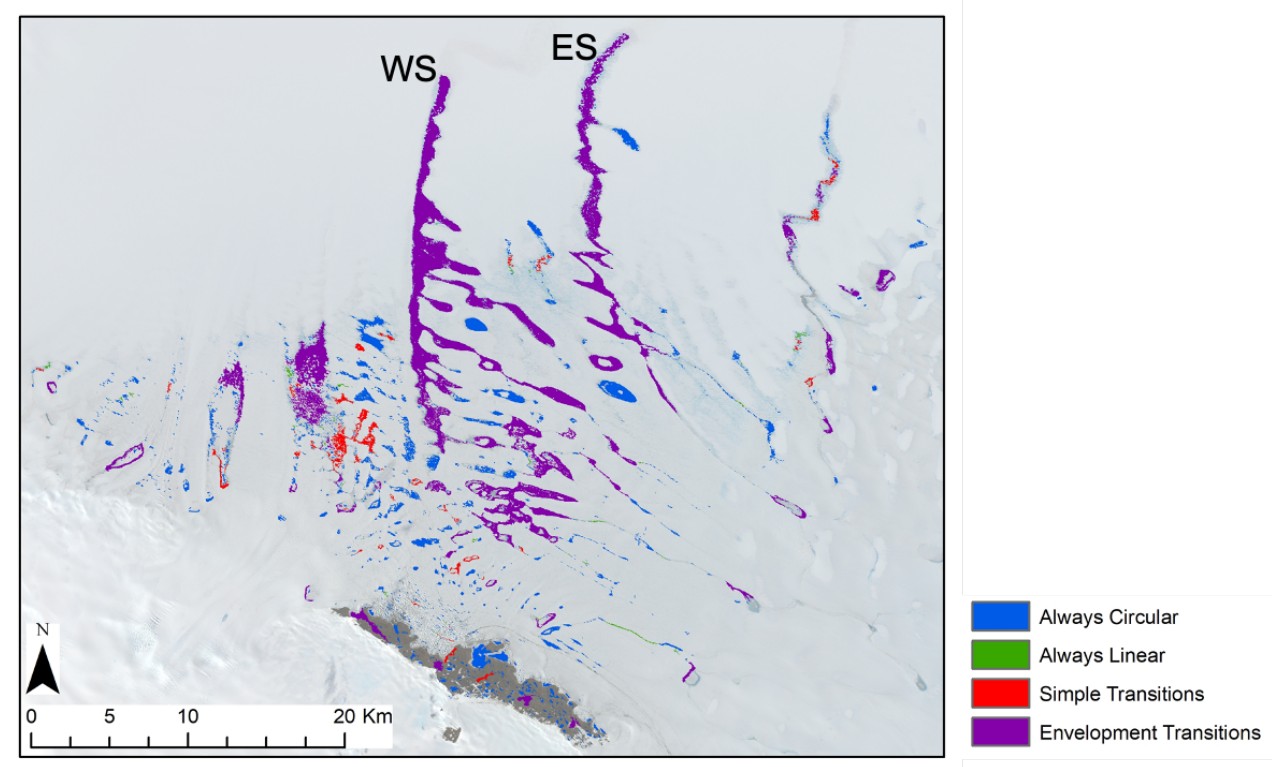


Figure 6: Maximum extent of all identified water bodies on the Nivlisen Ice Shelf for the 2016-2017
melt season, colour coded by water body type. 'WS' donates 'Western System', and 'ES' is
Eastern System. Base image aquired by Sentinel-2 on 26th January 2017.

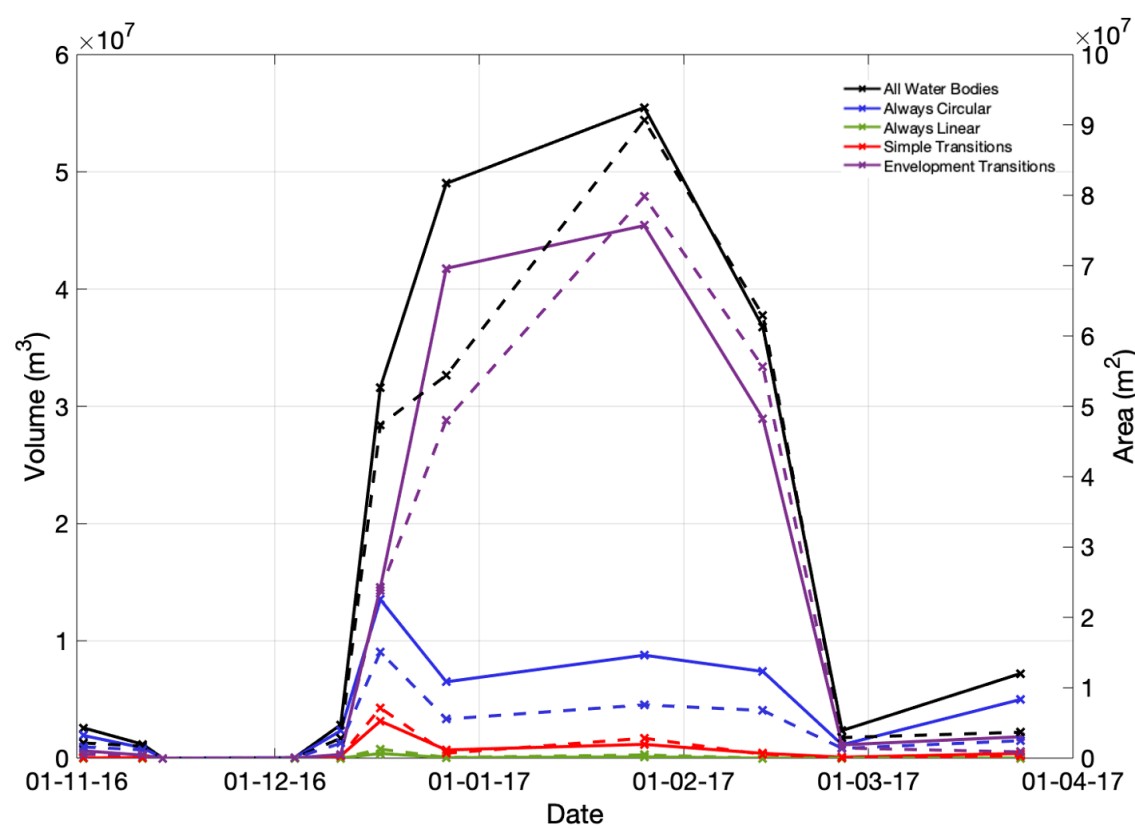

Figure 7: Time series of the total area and volume held in each water body category over the 2016-
2017 melt season on the Nivlisen Ice Shelf. Volumes are indicated by the solid lines, and areas by
the dashed lines.

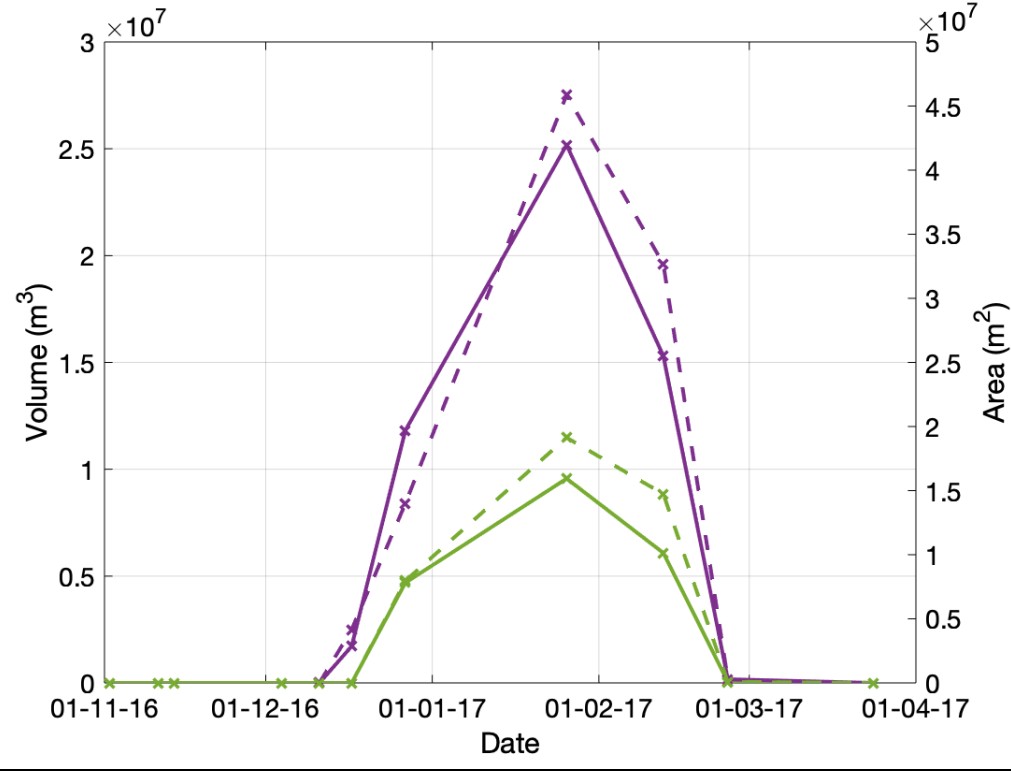



Figure 8: Time series showing the area (dashed line) and volume (solid line) of the WS (purple)
and ES (green).

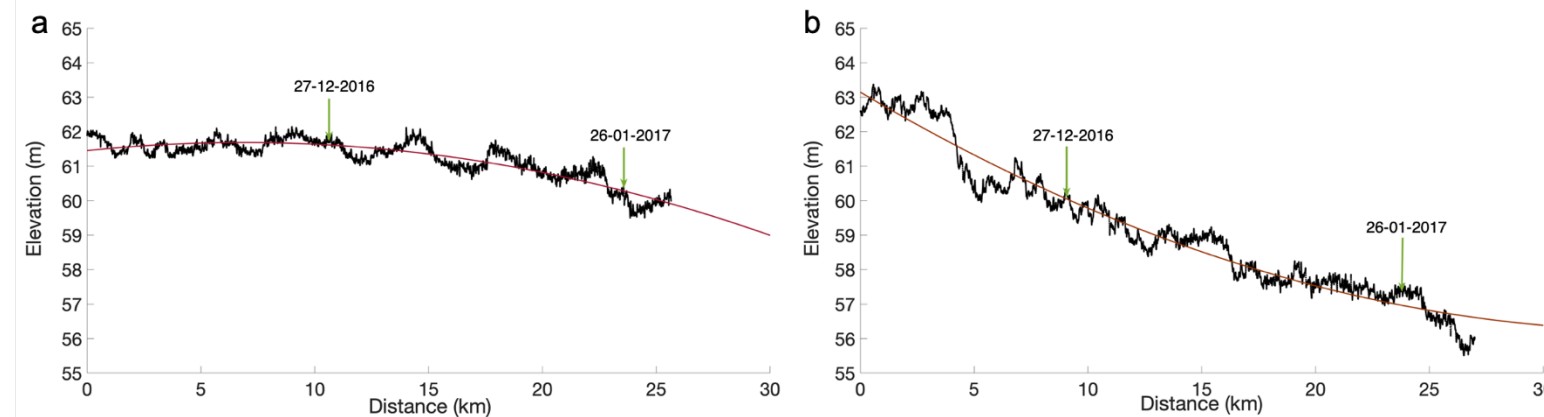


Figure 9: Elevation profiles for (a) the WS and (b) the ES. Quadratic trendlines are shown
in red. Data are extracted from REMA (Howat et al., 2019) and the path of data extraction
was guided using the maximum depth matrix of both the WS and ES over the full 2016-
2017 melt season (see Fig. S5). The labelled green arrows mark the down-ice extent of
each system on the 27th December 2016 and 26th January 2017.

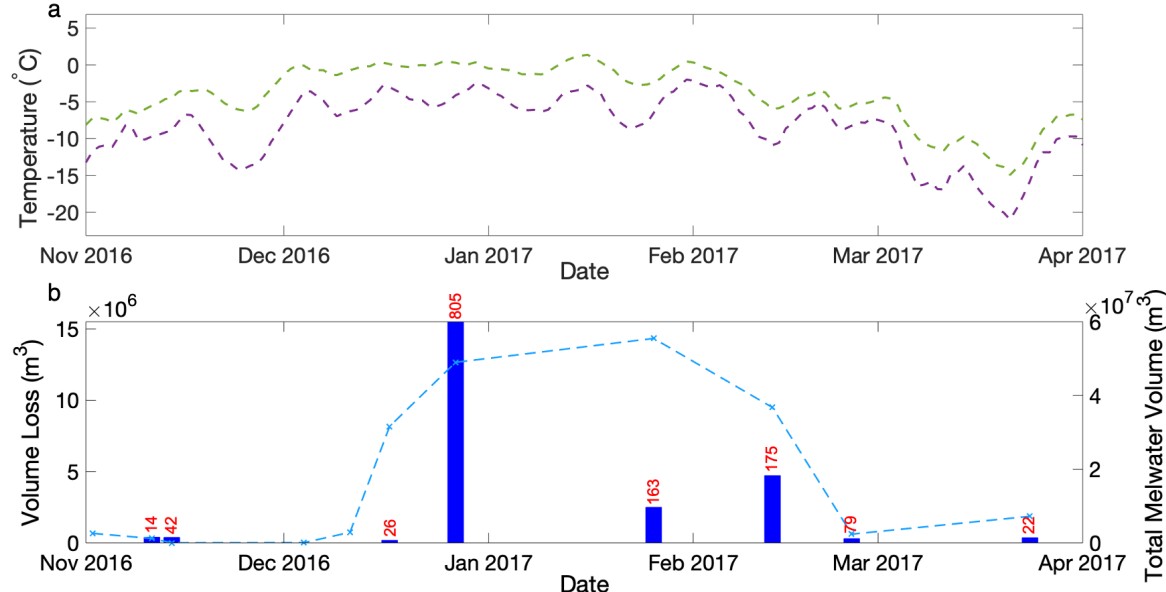

Figure 10: Meteorological context of circular lake loss events: a) The seven day moving average
of mean daily (purple line) and daily maximum (green line) near-surface air temperature from the
MetUM simulation for the period from November 2016 to April 2017 at the model point immediately
to the north of Schirmacheroasen. b) The total volume lost in 'loss events' by each image date
from water bodies in the 'always circular' category (blue bars) and the total combined water volume
(blue line). A loss event is defined as a > 80 % loss in water body volume through either lake
drainage or freeze-through.  The total number of loss events for each date is indicated above each
bar.

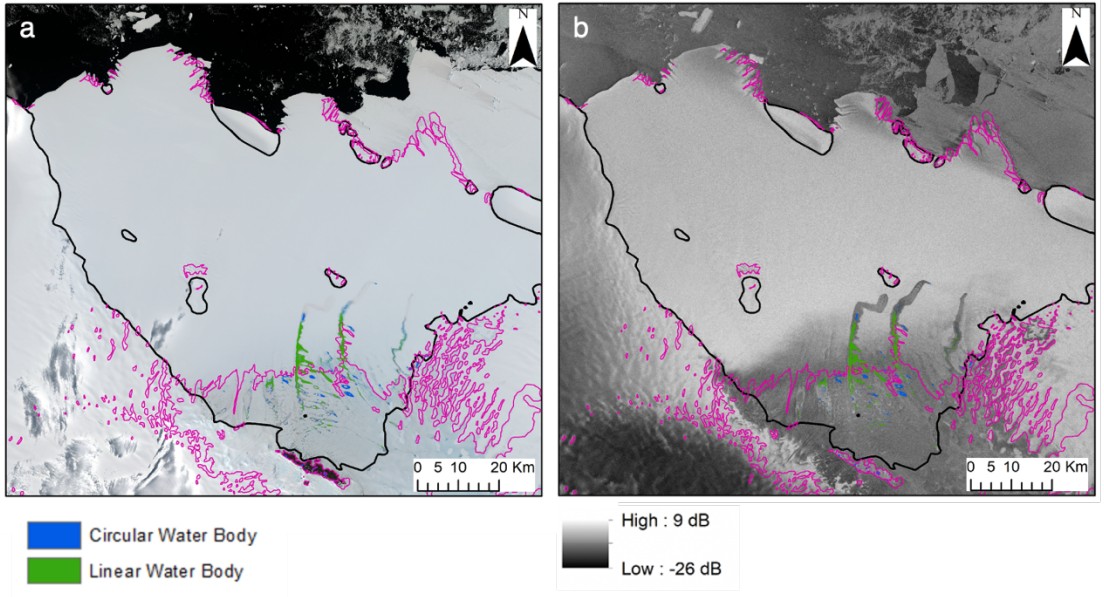

Figure 11: Comparison of optical imagery and radar imagery on 26th January 2017; a) is a
mosaicked Sentinel-2 image,  b) is a Sentinel-1 SAR image. Both a) and b) are overlain with the
blue ice extent (pink) and the mapped area of all linear and circular surface water bodies, based on
the FASTISh analysis of (a).

Table 1: Total area, total volume, and mean depth of all meltwater bodies on the Nivlisen Ice Shelf
on various dates in the 2016-2017 melt season.

| Date | Total Area (m$^2$) | Total Volume (m$^3$) | Mean Depth (m) | Max Depth (m) |
|---|---|---|---|---|
| 2nd November 2016 | 2.2 x 10$^6$ | 2.6 x 10$^6$ | 1.2 | 2.9 |
| 11th November 2016 | 1.7 x 10$^6$ | 1.2 x 10$^6$ | 0.7 | 2.6 |
| 14th November 2016 | 0.0 | 0.0 | 0.0 | 0.0 |
| 04th December 2016 | 4.4 x 10$^4$ | 4.0 x 10$^4$ | 0.9 | 3.1 |
| 11th December 2016 | 2.8 x 10$^6$ | 2.8 x 10$^6$ | 1.0 | 3.4 |
| 17th December 2016 | 4.7 x 10$^7$ | 3.2 x 10$^7$ | 0.7 | 3.1 |
| 27th December 2016 | 5.4 x 10$^7$ | 4.9 x 10$^7$ | 0.9 | 4.7 |
| 26th January 2017 | 9.1 x 10$^7$ | 5.5 x 10$^7$ | 0.6 | 3.3 |
| 13th February 2017 | 6.3 x 10$^7$ | 3.7 x 10$^7$ | 0.6 | 4.3 |
| 25th February 2017 | 2.9 x 10$^6$ | 2.4 x 10$^6$ | 0.8 | 3.0 |
| 24th March 2017 | 3.7 x 10$^6$ | 7.2 x 10$^6$ | 2.0 | 5.0 |



*Table 2: Maximum Area and Volume for each water body category on the Nivlisen Ice Shelf on*
*various dates in the 2016-2017 melt season.*

| | Maximum Area (m$^2$) | Maximum Volume (m$^3$) | Date of Maximum Volume | Date of Maximum Area |
|---|---|---|---|---|
| **All Water Bodies** | 9.1 x 10$^7$ | 5.5 x 10$^7$ | 26th January 2017 | 26th January 2017 |
| **Always Circular** | 1.5 x 10$^7$ | 1.4 x 10$^7$ | 17th December 2016 | 17th December 2016 |
| **Always Linear** | 1.3 x 10$^6$ | 3.9 x 10$^5$ | 17th December 2016 | 17th December 2016 |
| **Simple Transitions** | 3.2 x 10$^6$ | 3.2 x 10$^6$ | 17th December 2016 | 17th December 2016 |
| **Envelopment Transitions** | 8.0 x 10$^7$ | 4.5 x 10$^7$ | 26th January 2017 | 26th January 2017 |
