# Peer review of "Lateral meltwater transfer across an Antarctic ice shelf"

_The Cryosphere, 2019_

## Referee Comment (RC1) · Anonymous Referee #1 · 14 Feb 2020

**Lateral meltwater transfer across an Antarctic ice shelf**

Dell *et al.*

The Cryosphere

Anonymous referee

**General comments**:

This manuscript presents a quantitative analysis of meltwater distribution, storage and transfer on the Nivlisen Ice Shelf, Dronning Maud Land in East Antarctica during one melt season (2016-2017). The authors use a modified version of a previously published algorithm to classify meltwater into four different categories from Landsat-8 and Sentinel-2 imagery and track meltwater area and volume through the melt season.

Meltwater has been linked to the instability and collapse of Antarctic ice shelves, but detailed observations of the seasonal evolution of meltwater on ice shelves in Antarctica are lacking. Therefore, it is my view that the findings from this manuscript are of broad interest to the cryosphere community.

In general, the manuscript is well-written and clearly structured. The rationale for the study is well justified, including the specific choice of ice shelf. The quantification of meltwater area and volume stored and transferred on the ice shelf during one melt season builds upon previous work that has identified surface meltwater systems present on numerous Antarctic outlet glaciers and ice shelves (Langley et al., 2016; Kingslake et al., 2017; Stokes et al., 2019).

However, it is not immediately clear how the authors have adapted the previously published FASTER algorithm into the algorithm used here, FASTISh, and its novelty could be communicated more clearly. Its ability to differentiate between lakes and streams represents an important advance because it gives insight into the lateral transport of meltwater across ice shelves, not just the *in-situ* storage, but this needs to be made clearer in the Introduction. On a related note, I am slightly apprehensive about the binary classification of lake versus stream – what about elongate, linear lakes, which are common on Antarctic ice shelves?

There is a lack of discussion of the limitations associated with the methods used in Section 3, including classifying water body types, and applying the depth retrieval algorithm. The depth algorithm is known to only retrieve depths up to a certain point because rapid attenuation of red wavelengths means that only shallower lakes can be measured in this band (Pope et al., 2016), and makes several assumptions (see Sneed and Hamilton, 2011). The reader should be made aware of these limitations. Another point for the authors to consider, which deserves mention in the manuscript, is that this method will be unable to extract depth from lakes with floating ice cover (and hence will miss their deepest points).

The authors use surface and ground surface air temperature from one weather station to assess meltwater changes against temperature changes. However, this is only at a single point location, and I suggest the authors could compare reanalysis or regional climate model temperature data over the whole ice shelf, or at least over a broader area, at least to see if a similar pattern is shown through the melt season. Such models are spatially distributed and provide daily outputs. I believe that some regional climate models have outputs at 5.5 km resolution over parts of Dronning Maud Land. I leave it up to the authors to decide whether to include this additional analysis.

I would recommend that to strengthen the discussion in Section 5 the authors carry out flow routing analysis on a DEM in order to estimate the pathways in which surface meltwater would be routed. This would be particularly useful at end of third paragraph where the authors discuss the potential for excess meltwater export off the ice shelf.

The authors discuss the evolution of meltwater stored in the two largest meltwater systems. The meltwater volume loss from upstream lakes near the grounding line, and the gain in meltwater volume in these two large meltwater systems (WS and ES) through the melt season, should be more explicitly calculated in addition to the time series of total are and volume presented in Figure 6. This would help their discussion of lateral meltwater transfer across the shelf. A further optional suggestion is that the authors could include evidence that the firn contains shallow-sub-surface meltwater, e.g. visible as low backscatter in Sentinel 1 (Miles et al., 2017, Frontiers in Earth Science), which could aid their discussion of the likely processes responsible for meltwater transfer across the shelf.

Once the authors address these points, I can therefore recommend that this manuscript is suitable for publication in The Cryosphere.

**Specific comments**:

Line 30: Quantify how large (e.g. up to ~8/5 km long).

Line 55: Consider also citing Bevan et al. (2017) here, who present four boreholes on Larsen C in addition to that reported by Hubbard et al. (2016).

Line 90: See general comment above about highlighting the difference in algorithms.

Line 103: 'flow velocities of around 100 m $a^{-1}$' – is this at the grounding line or near the calving front?

Line 104: It would be helpful to label these regions on Figure 1, i.e. bare/blue ice area, Shirmacheroasen and the ice shelf itself. Red star is missing from Figure 1 inset.

Line 106: It would be useful to mention surface melt rates on the ice shelf in this section.

Line 117: Perhaps refer the reader to Figure 2 here.

Line 127: Perhaps explain why you have chosen to conduct your analysis on a single melt season, and why this specific melt season has been chosen. You state yourselves in line 82 the importance of tracking meltwater evolution through entire summer melt seasons, so why not do it here? If this was because your focus was on testing FASTish over a relatively small area and time window, make this clearer. Or was this purely based on imagery availability? Also, why not change 30th April 2017 to 24th March 2017, given this is the latest image you use. You also say you identified 12 scenes with 'minimal' cloud cover – did you select a specific % threshold?

Line 141: How well does this cloud mask perform compared to other cloud masks? And why was this method used but then you computed a cloud mask using a different method for Sentinel 2 scenes?

Line 166: As with Line 127, state the specific % cloud threshold used.

Line 184: I feel this is a bit vague – what was unsuccessful about this cloud masking approach? Were there false positives? Clouds not captured in the cloud mask? In addition, a little more

detail might be added about how individual masks were manually created in these cases – were polygons manually digitised in GIS?

Line 209: Explain what a two-dimensional connective threshold is.

Line 210: ≤ two pixels in total size, or in width? Same with Sentinel-2?

Line 216: Rather than qualitative depth assessments, I would say that NDWI has been primarily used previously to derive water body extents and area, so would delete this second part of the sentence.

Line 239: The description of how $A_d$ was calculated could be explained in slightly more detail.

Line 244: Did all images include deep water? What range of $R\infty$ values were used here? I believe Sneed and Hamilton (2011) conclude that optically-deep water is not required in every scene, and indeed Banwell et al. (2019) found a negligible difference of using a $R\infty$ value of zero. I am interested whether you explored different values to see what difference it made.

Line 250: Figure 2 mentions the removal of negative depths from the area and depth matrix, yet this is not discussed in-text. Where did negative depths originate from, was this in the cases with floating ice on lake surfaces?

Line 261: I think the solidity score needs to be explained in greater detail here, i.e. that the score can range from 0 to 1 and denotes the proportion of pixels within the convex hull of the water body, with streams or linear lakes having a score closer to 0, and perfectly circular lakes having a score of 1.

Line 276: Does FASTISh use the same minimum size threshold for tracking water bodies as in Williamson et al. (2018), i.e. 495 pixels (0.0495 km$^2$)? I couldn't see mention of this.

Line 278: I think this needs to be better highlighted that this is what sets FASTISh apart from the previous algorithm, FASTER (which did not track water body type).

Line 300: I think it needs to be made clear to the reader that a 'loss event' could either be a lake drainage or freeze-through. Given the previous sentence, one might assume this is just referring to lake drainage events only.

Line 309: I assume the single 2 m data strip was also REMA, but maybe best to clarify.

Line 314: See broader comments above.

Line 331: I suggest you include total area of meltwater bodies in each time-step (i.e. when discussing Fig 4A-D), as well as mean and maximum water depths.

Line 355: Interesting that the mean and maximum depth at the peak of the melt season is lower than in early December, near the start of the melt season? Have the authors considered why this may be?

Line 423: Is it not possible to distinguish whether a water volume loss event is freeze-through or surface drainage? This would help quantify specifically how much meltwater is transferred across the ice shelf.

Line 520: I would suggest moving the last sentence of this paragraph (explaining FAC) to here.

Line 546: How did you derive albedo over water-free distal areas of the ice shelf?

Line 572: Consider changing this to 'Around the peak of the melt season (26th January 2017)'

Line 576: Perhaps consider adding a sentence or two about the wider application of your findings. For example, your mapped lake extents could be used as constraints in surface hydrology modelling simulations?

Line 581: What about sharing your dataset publicly, i.e. lake extents containing attribute information of elevation, mean area, maximum area, mean depth, maximum depth.

Line 817: Looking at your study area extent, it seems odd to include a couple of lakes above the grounding line but not all, when there are visibly quite a few more within the blue ice area above the grounding line which fall outside your study area boundary. Given these are probably 'feeder lakes' which help route meltwater to lakes and channels on the ice shelf, it seems incomplete not to track seasonal changes in their area and volume as well.

Line 839: Are the white stripes across the figure artefacts? Consider changing the colour ramp to make values easier to read, and outlining lakes in black for visibility.

Line 852: Again, the colours make visibility difficult. I struggle to pick out areas in the 'always a stream' category

Line 877: I find this figure slightly confusing, though perhaps I have mis-interpreted it – why is one bar not shown per individual date studied in the 2016-2017 melt season (i.e. only 8 bars are shown, not 11). Clarify caption to show that loss events record either lake drainage *or* freeze-through.

**Technical corrections**

Line 23: inconsistent hyphenation of Landsat-8 and Sentinel-2 throughout the manuscript; please check throughout for consistency

Line 68: inconsistent hyphenation of sub-surface; please check throughout for consistency

Line 69: re-order citations chronologically; please check and modify throughout

Line 74: delete 'events'

Line 76: delete comma after 'front'

Line 81: change 'on' to 'in'

Line 102: is 'approximately' 123 km wide by 92 km long.

Line 161: 'change 'in' to 'into'

Line 175: Change 'give' to 'produce'

Line 201: Hyphenate 'water covered'

Line 218: Delete 'different'

Line 206: Change 'as' to 'because'

Line 220: Correct (Philpot, 1989) to Philpot (1989)

Line 294: Change 'track' to 'tracked'

Line 393: Maximum should be ~3 x$10^6$ m$^2$, not 2.5 x$10^7$?

Line 426: change 'by' to 'on' (as this is showing loss events on that particular date rather than cumulative events, if I have understood correctly)

Line 449: Change 'weather' to 'temperature fluctuations'

Line 467: New sentence after 'bodies'. 'We suggest this occurs when water ponds…'

Line 471: New sentence after 'dam'. 'Therefore, it is more likely to be the result…'

Line 497: hyphenate snow/firn-covered

Line 500: add 'firn' into 'available pore space'

Line 547: hyphenate 'large scale'

Line 863: Western and Eastern System are the wrong way round in the figure caption. Wrong dates in legends?

---

## Referee Comment (RC2) · Anonymous Referee #2 · 2 Mar 2020

This manuscript describes the evolution of surface meltwater on the Nivlisen Ice Shelf in Antarctica during the 2016-2017 melt season. The authors describe in detail their adaptation of the FAST algorithm for tracking water on ice shelves, and apply the method to two optical satellite datasets. They develop four categories of ice-shelf water bodies, and use them to discuss water transport across the Nivlisen Ice Shelf. Two main systems emerge on the ice shelf along linear surface depressions as the surface hydrology system transitions from isolated lakes to a connected network.

As explained in the introduction, ice-shelf surface hydrology is a critical component of ice-shelf stability, and thus important in predictions of ice-sheet mass loss. Being able to map and analyze the development of ice-shelf surface hydrology is important for understanding how future warming and subsequent ice-shelf melting will influence iceshelf behavior. There is a significant gap in our knowledge of water transport across ice shelves, and the goals of this manuscript are both timely and within the scope of TC. In general this manuscript is well written and should be published in TC with some revisions applied. The topic of ice-shelf meltwater transport is important, and the results of this study will certainly be of interest to the TC audience.

This manuscript builds on previously developed tools (water-tracking algorithms, band ratios, etc) and applies them to an ice-shelf whose surface hydrology has not to my knowledge been analyzed in detail. The authors have adapted these methods and tools to suit ice shelves, and combined this algorithm with previously validated methods of water volume retrieval from satellite imagery. However, the authors do not clarify why the FAST algorithm is not suitable for use on ice shelves and do not explain the differences between the FASTER algorithm and their new FASTISh.

The authors clearly demonstrate an evolving meltwater system on the Nivlisen Ice Shelf, and clearly describe why these observations are important for ice-shelf stability. However, the conclusions could use some clarification and possibly additional analysis. First of all, the difference between a meltwater lake and a stream is defined here only geometrically. Streams and rivers are defined by water flow, but the definition used here does not include a water flow condition. Rivers and streams described here could technically be linear lakes that fill up from percolation through the firn. The authors seem to support this, since they describe two mechanisms for water transport 1) movement through firn and 2) movement via overtopping lakes, but then claim there is no evidence of overtopping lakes. There is not sufficient data or analysis to confirm water is flowing in these features precisely because water transport in the firn cannot be tracked from optical satellite imagery. Satellite radar could be used to track melt in the upper firn layers and might provide insights into exactly how water is moving. Additionally, the authors have not explained where firn is located on the ice shelf or how it is detected. I think it was visually identified. That should be briefly described and indicated on one or more maps (along with blue ice areas). The connection between
temperature and surface hydrology development could be enhanced by including data from a climate model that covers the entire ice shelf as opposed to a single weather station. The development of air and ground temperatures could be tracked through space and time and then be compared to the development of the surface hydrology system that the authors already quantified.

The general structure of the manuscript could be improved by including some of the more important conclusions and implications in the abstract and introduction. For example, the comparison to the surface lakes of Larsen B in the discussion could be highlighted to provide some motivation for the study, as opposed to limiting the goal of the study to developing an algorithm and tracking water on one ice shelf. Some more context for the Nivlisen Ice Shelf would be helpful in Section 2. In terms of the methods, providing context for the FASTISh algorithm would go a long way. A lot of time is spent in the beginning discussing a method that is at this point standard in ice-shelf/ice-sheet hydrology, and it's not clear how FASTISh differs from FAST.

Figures:

Titles would help the reader interpret figures more easily.

Figure 1. Add coordinates to Nivlisen map, include location of weather station. Annotate blue ice areas, water bodies and firn extent. The caption refers to a red star to locate the Nivlisen Ice Shelf, but I don't see it. Also, is this image mosaicked? From the description of data preparation, it seems like the entire ice shelf cannot be captured by a single Landsat scene, and that pairs of scenes were mosaicked.

Figure 2. This is a nice workflow diagram. One very detail-oriented comment: It is a little confusing to have "Cloud Mask" in an input, and then the next line be the "Apply Cloud Mask" processes in the same workflow. From my understanding of the workflow, cloud masks are generated from a separate workflow followed in Google Earth Engine and are then an output of that workflow. Maybe delete the "Cloud Mask" input from the top.
Figure 3. Include the paths of the DEM that were extracted and shown in Figure 8. Consider combining this figure with Figure 8 and maybe Figure 7. It would be nice to have an "elevation figure" to synthesize everything.

Figure 4. This shows the water transport really nicely. Consider having the scale bar on only one image so that the eye can focus more on the water. Also, consider making the color of lakes and streams more distinct- it is hard to distinguish in print and only slightly easier on screen. Also, the text references streams a and b on line 343, but I think you should keep names and labels consistent. Label them here, since it is the first time we see the hydrologic system. Maybe add labels to row D.

Figure 5. Great figure! Move the WS and ES labels so they don't overlap the nice masks. Is the basemap Landsat? Include that in the caption. Is it necessary to include a category for water that isn't mapped if the background shows a satellite image? It makes me want to look for what features with a gray outline.

Figure 6. Get rid of white space after 01-04-17 since nothing is plotted there. Actually, if I'm correct in this interpretation, it seems like this figure shows a period of water deepening– in the All Water Bodies and Envelopment Transitions categories, the volume increases at a greater rate than the area from  $\sim$  01-12-16 to 01-02-17.

Figure 7. This figure basically tells me that water flows downhill. This is very encouraging, but I think that's communicated in the combination of Figures 3 and 4.

Č Specific comments:

Line 1: change "stored" to "can form"

Line 31: I don't think it's clear that water transfer is "facilitated by two large surface streams." In fact, the discussion seems to indicate lateral transport of water through firn.

Line 80: I agree it is important to assess the variability of ice-shelf surface hydrology over the course of a melt season, but the year-to-year change is also critical. As just

TCD
one example, the Nansen River does not form every year (Bell et al. 2017). If meltwater transport is such an important part of ice-shelf stability, then its variable behavior must be important, too. What if water didn't move across Nivlisen one year?

Line 92: Why is it necessary to produce FASTISh? It would be very helpful to include a description of FAST, highlighting its strengths and weaknesses.

Line 98: In general I would like more description of the ice shelf here. A discussion of the firn would be useful.

Line 103: Be more specific about where the ice thickness reaches 700 m. Is 700 m a grounding line thickness? Taking a look at Horwath et al., referred to in the text, it looks like it might be.

Line 105: It would be helpful to include a reference to Fig. 1 here with some annotations provided.

Line 110 (Study Area in general): Discuss meltwater features in more detail than "meltwater features show changes over time." Kinglsake et al. 2015 specifically say that lakes drain on the Nivlisen Ice Shelf. You may disagree with the classification of the meltwater identified by Kingslake et al. 2015 – the images there look like what you call an "envelope transition." However, "changes over time" doesn't communicate enough information to the reader. This description could include basic information from your analysis, too. Knowing up front how large the extent of melting is compared to the total area of the ice shelf, for example, would be helpful in orienting the reader. Also including some basic climate information would be good, especially since you talk about temperature data later.

Line 139: Converting from DN to TOA values is important for analyzing spectra anywhere, not just at high latitudes.

Line 156: I appreciate that clipping each scene to the same extent is a required step in the algorithm, but I don't understand the reason. Although you say "when com-
paring images," but you can compare scenes that have overlapping areas but not the same extent. This makes me think there is some other requirement that you haven't mentioned.

Line 168: It might be good to describe bands in terms of their spectral range rather than color, since you are using imagery from two different platforms. Clearly this is not critical, just a thought.

Line 171: The division by a "quantification value" comes out of nowhere. Is there a paper you could cite that explains this as a part of data processing? I would delete "expected" since it can imply that something is wrong with Sentinel-2, when it seems like it's just a convention of how they deliver their data. I haven't worked much with Sentinel-2, however, so if this really is an unexpected discrepancy then ignore my suggestion.

Line 178: Is the thresholding approach the same as before?

Line 179: How can a value exceed 10,000 if all bands were divided by 10,000?

Line 233: While I know it is correct to say that errors in lake depth estimates have been calculated as zero (Pope et al. 2016), I think it is misleading to have this be the only mention of error about this method. To a reader who isn't already familiar with this method, the claim that error is zero will be shocking, and probably lead to skepticism. I think that if you are going to describe the method to help the reader understand what you did, it is important to give a little bit more detail about the sensitivity to A and g, the dependence of the band chosen, and that physical qualities of the system are known (water column attenuation, surface roughess of the lake, etc.)

Line 265: I'd really like to see a figure of the thresholds and different water body shapes. I think it would help me understand the categories you introduce in section 3.5.

Line 286: There should definitely be a figure for this. It could be Figure 5, but I would almost prefer a zoomed-in high detail of an area with one of each type.
Line 308: Is the figure for this Fig. 7? If so, refer to it. But also, be clear about which water bodies. I was expecting an elevation profile with different colors representing your four water body categories, whereas Fig. 7 shows the Eastern and Western Systems.

Section 3.7: Plot this on a map-I can't figure out where it is! Also I think a single point of temperature is insufficient for this analysis. It would be useful to see modeled (maybe RACMO) temperature and energy balance across the extent of the surface hydrology system.

Line 326 and Section 4.1: Give a general synthesis. Instead of only having four paragraphs that describe the observations, tell the reader the overall trend. I can't really grab on to a take-away message the way it is written.

Line 328-29: Ice shelves are flat places- what does relatively flat mean and how do you know how thin the winter snow cover is?

Line 343-344: Refer to Fig. 5.

Line 353: Again, how do you know the extent of the firn? If it is visually identified, consider digitizing it. I'd be curious to see how the firn extent changes as the system evolves, especially given your discussion section.

Line 357-358: I am now getting confused between lakes and streams- are the two "large streams" just linear lakes? Your distinction between the two is only geometric. It is not clear that water flows in these water bodies the way water flows in streams. I am convinced water is transported, but I don't think there is clear evidence for how. Except that you later say you have no evidence for water overtopping lakes, which makes me think you are concluding most water moves through firn.

Line 385-387: The FASTISh algorithm itself does not produce a time series of water body properties- it only does you have used it on a time series of images.

Line 392-397: While you refer to Fig. 7 for this, that figure only shows migration to lower elevations, not lower latitudes. I am not sure what trends you refer to in section
4.1 – rewriting that section to include a general point would help. Then refer to it here by saying "In line with the XYZ trends described in section 4.1...."

Line 399: I got confused between the Eastern System and Western System and the water body categories. When I looked at Fig. 7 I was expecting to see the four categories, since the title of the section is "Tracking Individual Water Bodies." Why have you not done this? Is it because most of the water is in the ES and WS? It would make sense to focus on these two, given that there is also migration of both systems. But I think that needs to be clarified in the text. It is even more confusing in the rest of the paragraph because on line 404 you say "most of the surface water" and it's not clear if you mean "most of the surface water in each meltwater system" (and I don't even know which meltwater system it is, east or west?) or if you mean "most of the surface water on the ice shelf".

Line 418-419: I'm not sure this sentence matters. What is rhythmic variance? What do these properties mean for the surface water?

Line 448: You say surface water area and volume correspond with rising temperatures, but Fig. 9 shows volume loss is associated with temperature increase, and only in the category of "always a lake" when most of the water is contained in the ES and WS. It would be helpful to include a figure showing what is claimed on line 453 "As temperatures rise and surface water bodies increase in area and volume...." It seems like the point you are trying to make in this section is that temperature controls the evolution of the system, so showing the decreasing lakes with increasing envelopment transition might make that clearer. Although later it seems like you are claiming the opposite.

Line 463: How do you know water is flowing along linear depressions as opposed to those depressions being filled from the sides due to transport in the firn? Especially since you mention on line 470-472 that the growth of ES and WS are more likely to be fed by increased meltwater production and movement through firn. In general, I think
you argue well for meltwater transport through firn. I would highlight that, but it would require a little more discussion and analysis of how you track firn on the surface.

Line 464-468: You have not demonstrated that the ES and WS are "fed by smaller surface lakes and streams both above and below the grounding line."

Line 470-472: Is it really possible to infer this when there are no images between Dec. 27th and Jan 26th? I assume that evidence of a "lip" would be the development of a stream forming out of a lake. How do you know this didn't happen? Also, refer to Fig. 4 here.

Line 480-482: You claim it is possible that meltwater could be transported to the front of the ice shelf in the future. That would require water to organize in a river that can flow off the ice-shelf edge (unless there is some meltwater evacuation through firn that I don't know about, but that's another can of worms). You could potentially demonstrate this by assuming at some point the firn will saturate and water will follow surface depressions (you have already argued that this is happening). You could perform a water routing analysis on the full DEM, or at least see if you can visually identify linear surface depressions out to the edge of the shelf.

Line 488-501: "The relatively extensive snow and firn cover ...likely prevents the development of...meltwater...on the Larsen B." This is a very important distinction that I think you should highlight (maybe include it in the abstract and allude to it in the introduction?). This is definitely an important conclusion, because it defines a type of meltwater type and evolution that stands in contrast to the "ponds break ice shelves" style of meltwater without needing to remove water.

Line 515-518: Earlier you said that the transfer of water was \*not\* due to overtopping of surface lakes.

Line 508-526: It seems like now you are saying temperature does not control the evolution of the surface hydrology system. Earlier it seemed like you were saying the
opposite.

Line 530-552: What are the implications for the stability of Nivlisen? Could you tie your analysis more explicitly to our general understanding of ice-shelf stability? Your refreezing calculation is really interesting, but has nothing to do with the rest of the paper. It makes me again want to know more about firn on this ice shelf.

Line 558-576: I would lead with the conclusions you reach about the ice shelf, not with a description of your method, and also include a summary of the implications of your analysis.

Line 570: Again, how do you know there is a thin snow band and firn pack?

Č Technical comments

Line 20: hyphenate "ice-shelf"

Line 59: hyphenate "ice-shelf"

Line 76: hyphenate "ice-shelf"

Line 210: change "two" to "2" since it follows a mathematical symbol, even though it is a number less than ten.

Line 294: "identified and tracked"

Line 306-307: Spell out acronyms when you use them for the first time REMA and DEM.

Line 548: hyphenate "ice-shelf"

---

## Referee Comment (RC3) · Anonymous Referee #2 · 2 Mar 2020

I forgot to add that I really enjoyed reading this paper and it gave me a lot to think about.

-Reviewer 2

---

## Author Comment (AC1) · 23 Apr 2020

Thank you for your positive review, please find attached our responses.

[Figure]

Rebecca Dell
Scott Polar Research Institute
University of Cambridge
Lensfield Road
Cambridge CB2 1ER
United Kingdom

Tel: +44 (0) 7925618105
Email: rld46@cam.ac.uk

30th March 2020

Dear Dr Whitehouse,

RE: Response to Reviewer Comments

We have received two reviews of our manuscript entitled 'Lateral meltwater transfer across an Antarctic ice shelf (tc-2019-316). Overall, both reviews were supportive of the paper and its relevance to the wider research community.

Both reviewer's comments have been extremely helpful in further improving the manuscript. The specific changes made can be found in our responses below. Referee comments are italicised in *blue*, our responses are in black. Please also find attached a revised manuscript with all tracked changes. When referring to page numbers in the below text, these will be in line with the page numbers on the 'tracked changes' manuscript.

Thank you for taking your time to consider our revised manuscript, we hope it is now acceptable for publication.

Sincerely,

*R. Dell*

Rebecca Dell and co-authors

**Fig. 1.**

---

## Author Comment (AC2) · 23 Apr 2020

Rebecca Dell
Scott Polar Research Institute
University of Cambridge
Lensfield Road
Cambridge CB2 1ER
United Kingdom

Tel: +44 (0) 7925618105
Email: rld46@cam.ac.uk

30th March 2020

Dear Dr Whitehouse,

RE: Response to Reviewer Comments

We have received two reviews of our manuscript entitled 'Lateral meltwater transfer across an Antarctic ice shelf (tc-2019-316). Overall, both reviews were supportive of the paper and its relevance to the wider research community.

Both reviewer's comments have been extremely helpful in further improving the manuscript. The specific changes made can be found in our responses below. Referee comments are italicised in blue, our responses are in black. Please also find attached a revised manuscript with all tracked changes. When referring to page numbers in the below text, these will be in line with the page numbers on the 'tracked changes' manuscript.

Thank you for taking your time to consider our revised manuscript, we hope it is now acceptable for publication.

Sincerely,

R. Dell

Rebecca Dell and co-authors

**Response to Reviewer Comments – Reviewer 1**

**Overall Comments**

*This manuscript presents a quantitative analysis of meltwater distribution, storage and transfer on the Nivlisen Ice Shelf, Dronning Maud Land in East Antarctica during one melt season (2016-2017). The authors use a modified version of a previously published algorithm to classify meltwater into four different categories from Landsat-8 and Sentinel-2 imagery and track meltwater area and volume through the melt season. Meltwater has been linked to the instability and collapse of Antarctic ice shelves, but detailed observations of the seasonal evolution of meltwater on ice shelves in Antarctica are lacking. Therefore, it is my view that the findings from this manuscript are of broad interest to the cryosphere community. In general, the manuscript is well-written and clearly structured and the rationale for the study is well justified. The quantification of meltwater area and volume stored and transferred on the ice shelf during one melt season builds upon previous work that has identified surface meltwater systems present on numerous Antarctic outlet glaciers and ice shelves (Langley et al., 2016; Kingslake et al., 2017; Stokes et al., 2019).*

We thank Reviewer 1 for this positive overview and their further constructive review.

**Major Comments**

*It is not immediately clear how the authors have adapted the previously published FASTER algorithm into the algorithm used here, FASTISh, and its novelty could be communicated more clearly.*

And also minor comments: *Line 90: See general comment above about highlighting the difference in algorithms.*

*Line 278: I think this needs to be better highlighted that this is what sets FASTISh apart from the previous algorithm, FASTER (which did not track water body type).*

We agree with these comments, and have made the differences between the FASTER and FASTISh algorithms clearer (lines 123-133), we have also added a paragraph justifying the need for these adaptations (lines 100-117).

*Its ability to differentiate between lakes and streams represents an important advance because it gives insight into the lateral transport of meltwater across ice shelves, not just the in-situ storage, but this needs to be made clearer in the Introduction.*

We have now made this difference between FASTER and FASTISh clearer in introduction (lines 123-133) and also in the abstract (lines 26-27).

*On a related note, I am slightly apprehensive about the binary classification of lake versus stream – what about elongate, linear lakes, which are common on Antarctic ice shelves?*

Based on our current level of analysis, we now agree that it is not possible to confidently categorise water bodies as lakes or streams. We have therefore re-named our categories to purely reflect the geometry of the water bodies, which are now classified as either circular or linear, regardless of whether they contain ponded or flowing water. These adjustments have been made where appropriate throughout the manuscript.

*There is a lack of discussion of the limitations associated with the methods used in Section 3, including classifying water body types, and applying the depth retrieval algorithm. The depth algorithm is known to only retrieve depths up to a certain point because rapid attenuation of red wavelengths means that only shallower lakes can be measured in this band (Pope et al., 2016), and makes several assumptions (see Sneed and Hamilton, 2011). The reader should be made aware of these limitations.*

As above, we have changed the water body type definitions from 'lakes' and 'streams' to 'circular' and 'linear' features; this should remove limitations as the definition of each water body is now purely geometric. We have incorporated a comprehensive review of the limitations and assumptions of the depth algorithm into Section 3 (lines 284-290).

*Another point for the authors to consider, which deserves mention in the manuscript, is that this method will be unable to extract depth from lakes with floating ice cover (and hence will miss their deepest points).*

We have clarified that water bodies with a floating ice cover effectively had their ice cover 'removed' and 'filled in' with water so the areas of these lakes are as accurate as possible. Pixels that had been 'filled in' due to ice cover were then prescribed the mean depth of their respective water bodies:

On lines 264-266, we have added: "To ensure that pixels with floating ice cover were still included in the analysis, we then used the 'imfill' function within MATLAB to classify any 'dry' pixels situated within a water body as water."

And on lines 311-313: "Pixels that had a floating ice cover and had been filled (see section 3.2) were assigned the mean depth of their respective water bodies, as it is not possible to calculate the depth of a pixel with an ice cover."

*The authors use surface and ground surface air temperature from one weather station to assess meltwater changes against temperature changes. However, this is only at a single point location, and I suggest the authors could compare reanalysis or regional climate model temperature data over the whole ice shelf, or at least over a broader area, at least to see if a similar pattern is shown through the melt season. Such models are spatially distributed and provide daily outputs. I believe that some regional climate models have outputs at 5.5 km resolution over parts of Dronning*

*Maud Land. I leave it up to the authors to decide whether to include this additional analysis.*

*And minor comment:* Line 314: See broader comments above.

We agree that this would be better, and we have therefore analysed data from "from an atmosphere-only regional climate CORDEX (COordinated Regional climate Downscaling Experiment) simulation of Antarctica using the limited-area configuration of Version 11.1 of the UK Met Office Unified Model (MetUM) for the period 2016-2017." Please see section 3.7 (lines 377-401) for a full explanation.

*I would recommend that to strengthen the discussion in Section 5 the authors carry out flow routing analysis on a DEM in order to estimate the pathways in which surface meltwater would be routed. This would be particularly useful at end of third paragraph where the authors discuss the potential for excess meltwater export off the ice shelf.*

We agree that this would be useful, and we attempted to carry out this analysis prior to initial submission. Unfortunately, the REMA data contains too many significant elevation errors and data gaps, meaning that flow routing analysis is extremely hard. 'Tidying up' the REMA data would be a large project in itself, and is something we think is beyond the scope of this current study.

*The authors discuss the evolution of meltwater stored in the two largest meltwater systems. The meltwater volume loss from upstream lakes near the grounding line, and the gain in meltwater volume in these two large meltwater systems (WS and ES) through the melt season, should be more explicitly calculated in addition to the time series of total area and volume presented in Figure 6. This would help their discussion of lateral meltwater transfer across the shelf. A further optional suggestion is that the authors could include evidence that the firn contains shallow-sub- surface meltwater, e.g. visible as low backscatter in Sentinel 1 (Miles et al., 2017, Frontiers in Earth Science), which could aid their discussion of the likely processes responsible for meltwater transfer across the shelf.*

We have included a new figure (Fig.8), which shows the volume and area of both the WS and ES throughout the melt season. This figure is discussed further in lines 498-502.

We have also included an additional new figure (Fig.11), which show an optical and a SAR image, this adds to our discussion (lines 583-597), and provides evidence of sub-surface meltwater storage within the firn.

**Minor Comments**

Line 30: Quantify how large (e.g. up to ~8/5 km long).

Corrected (Line 29)

Line 55: Consider also citing Bevan et al. (2017) here, who present four boreholes on Larsen C in addition to that reported by Hubbard et al. (2016).

Corrected (Line 58)

Line 103: 'flow velocities of around 100 m a$^{-1}$' – is this at the grounding line or near the calving front?

We realised that our previous sentence was vague. We have reworded this sentence to say the following (Lines 141-143): "Ice thickness ranges from 150 m at the calving front to ~ 700 m towards the ice shelf's grounding line in the southeast, and it exhibits flow velocities of around 20 m a$^{-1}$ to 130 m a$^{-1}$ (Horwath et al., 2006)."

We have not commented on the spatial arrangement of these flow velocities, as we see no obvious spatial gradient in Figure 7 of Horwath et al. (2006) (page 23):

[Figure]

**Fig. 7.** Horizontal velocity field of Nivlisen derived from interferometric SAR processing. Black lines are flowlines deduced from the amplitude image and used to define the flow direction. Arrows show the GPS measurements used for the calibration.

Line 104: It would be helpful to label these regions on Figure 1, i.e. bare/blue ice area, Shirmacheroasen and the ice shelf itself. Red star is missing from Figure 1 inset.

Corrected (Figure 1)

Line 106: It would be useful to mention surface melt rates on the ice shelf in this section.

To the best of our knowledge, no surface melt rates have yet been measured on the Nivlisen Ice Shelf.

Corrected (Line 164)

Line 127: Perhaps explain why you have chosen to conduct your analysis on a single melt season, and why this specific melt season has been chosen. You state yourselves in line 82 the importance of tracking meltwater evolution through entire summer melt seasons, so why not do it here? If this was because your focus was on testing FASTish over a relatively small area and time window, make this clearer. Or was this purely based on imagery availability?

We have justified this in lines 131-133 (Section 1) as we feel it is better placed here: "We then apply the FASTISh algorithm to the Nivlisen Ice Shelf, Antarctica, for the 2016-2017 melt season; the first full melt season to have data coverage over the ice shelf from both Landsat 8 and Sentinel-2."

Also, why not change 30th April 2017 to 24th March 2017, given this is the latest image you use.

Corrected (Line 173)

You also say you identified 12 scenes with 'minimal' cloud cover – did you select a specific % threshold?

And Line 166: As with Line 127, state the specific % cloud threshold used.

We visually picked images, as using a % threshold doesn't provide information on where in an image scene the cloud cover is. If there is 30 % cloud cover but off the ice shelf, we could still use the image. Whereas if there is 10 % cloud cover but over the main melt area, the image is not useful.

Line 141: How well does this cloud mask perform compared to other cloud masks? And why was this method used but then you computed a cloud mask using a different method for Sentinel 2 scenes?

We have clarified this in lines 202-204: "We found the 'simple cloud score algorithm' to be the most effective cloud masking method for Landsat 8 images, as it assesses each pixel using multiple criteria, making it more effective than any single band threshold."

And lines 229-232: "As the simple cloud score algorithm had not been adapted for application to S2 imagery at the time of writing, we computed a cloud mask for each image using a thresholding approach, whereby pixels were categorised as cloudy if the SWIR band value was > 10,000."

Line 184: I feel this is a bit vague – what was unsuccessful about this cloud masking approach? Were there false positives? Clouds not captured in the cloud mask? In addition, a little more detail might be added about how individual masks were manually created in these cases – were polygons manually digitised in GIS?

We have clarified these points in lines 235-238: "On two image dates (14th November 2016 and 25th February 2017), this cloud masking approach was not entirely

successful as not all clouds were fully masked. Additional individual masks were manually digitised in ArcGIS to ensure all clouds were masked for these images."

Line 209: Explain what a two-dimensional connective threshold is. Line 210: ≤ two pixels in total size, or in width? Same with Sentinel-2?

We have clarified these points in lines 259-264: "Having applied a 0.25 NDWI$_{ice}$ threshold to each image, the resulting water masks were filtered using a two-dimensional 8-connected threshold (i.e. grouping pixels if they were connected by their edges or corners) to identify each individual water body. Water bodies consisting of ≤ 2 pixels (Landsat 8) and ≤ 18 pixels (Sentinel-2), were removed to ensure only water bodies with an area ≥ 1,800 m$^2$ were assessed further."

Line 216: Rather than qualitative depth assessments, I would say that NDWI has been primarily used previously to derive water body extents and area, so would delete this second part of the sentence.

We agree, and have actually removed the beginning of this paragraph, as it wasn't very succinct (Line 270).

Line 239: The description of how Ad was calculated could be explained in slightly more detail.

Clarified in lines 296-298.

Line 244: Did all images include deep water? What range of R∞ values were used here? I believe Sneed and Hamilton (2011) conclude that optically-deep water is not required in every scene, and indeed Banwell et al. (2019) found a negligible difference of using a R∞ value of zero. I am interested whether you explored different values to see what difference it made.

Corrected (Lines 307-308): "For images that did not contain optically deep water, the R$_\infty$ value was set to 0 (Banwell et al., 2019)."

Line 250: Figure 2 mentions the removal of negative depths from the area and depth matrix, yet this is not discussed in-text. Where did negative depths originate from, was this in the cases with floating ice on lake surfaces?

Clarified In lines 301-304: "In very rare cases, wet pixels within a water body could have a reflectance higher than the calculated $A_d$ value, leading to negative water depths. All such pixels were removed from the area and depth matrix (Fig. 2)."

Line 261: I think the solidity score needs to be explained in greater detail here, i.e. that the score can range from 0 to 1 and denotes the proportion of pixels within the convex hull of the water body, with streams or linear lakes having a score closer to 0, and perfectly circular lakes having a score of 1.

Further clarification added (lines 322-329).

Line 276: Does FASTISh use the same minimum size threshold for tracking water bodies as in Williamson et al. (2018), i.e. 495 pixels (0.0495 km$^2$)? I couldn't see mention of this.

We track all the water bodies we identify. The threshold we apply for categorising water in a water body is already stated in Section 3.2 (lines 262-264).

Line 300: I think it needs to be made clear to the reader that a 'loss event' could either be a lake drainage or freeze-through. Given the previous sentence, one might assume this is just referring to lake drainage events only.

Sentence modified as suggested (lines 514-519): "Therefore, we used the calculated volume time series to identify water bodies in the 'always circular' category that lost > 80 % of their maximum volume over the full melt season, through either drainage or freeze through. We focus solely on the 'always circular' category to better understand the local loss of surface melt water in seemingly isolated and stationary water bodies. These events are referred to as 'loss events' hereafter." Please note we have moved this section into the results (Section 4.2.3) to improve the flow of the text.

Line 371: I assume the single 2 m data strip was also REMA, but maybe best to clarify.

Corrected (Line 364)

Line 331: I suggest you include total area of meltwater bodies in each time-step (i.e. when discussing Fig 4A-D), as well as mean and maximum water depths.

We agree, and also included total volume. These results are also shown in a new table (Table 1).

Line 355: Interesting that the mean and maximum depth at the peak of the melt season is lower than in early December, near the start of the melt season? Have the authors considered why this may be?

Yes, we have added an explanation for this in lines 480-486: "Between 17[th] December 2016 and 27[th] December 2016 'all water bodies' are characterised by 'deepening', as their total volume increases at a greater rate than their total area, and their mean depth increases (Tables 1 and 2, Fig. 7). Whereas between the 27[th] December and the 26[th] January, 'all water bodies' are characterised by 'spreading', as their total area increases at a faster rate than their total volume, and the mean water body depth decreases (Tables 1 and 2, Fig. 7)."

Line 423: Is it not possible to distinguish whether a water volume loss event is freeze-through or surface drainage? This would help quantify specifically how much meltwater is transferred across the ice shelf.

The temporal resolution of our data set (which is a factor of available cloud free imagery) is unfortunately not good enough to allow us to identify drainage events.

Line 520: I would suggest moving the last sentence of this paragraph (explaining FAC) to here.

We have changed our wording slightly, but the discussion around FAC is now at the end of the paragraph (lines 573-575).

Line 546: How did you derive albedo over water-free distal areas of the ice shelf?

We digitised a polygon over a water-free distal part of the ice shelf, and calculated the average albedo of the pixels within this polygon (lines 666-667).

Line 572: Consider changing this to 'Around the peak of the melt season (26[th] January 2017)'

Corrected (line 691).

Line 576: Perhaps consider adding a sentence or two about the wider application of your findings. For example, your mapped lake extents could be used as constraints in surface hydrology modelling simulations?

Lines 696-697: "Our findings could be useful in comparing to IceSat 2 derived lake depths, in addition to constraining future ice-shelf surface hydrology models."

Line 581: What about sharing your dataset publicly, i.e. lake extents containing attribute information of elevation, mean area, maximum area, mean depth, maximum depth.

Upon publication, we will upload our scripts to the Apollo Repository (Lines 701-703).

Line 817: Looking at your study area extent, it seems odd to include a couple of lakes above the grounding line but not all, when there are visibly quite a few more within the blue ice area above the grounding line which fall outside your study area boundary. Given these are probably 'feeder lakes' which help route meltwater to lakes and channels on the ice shelf, it seems incomplete not to track seasonal changes in their area and volume as well.

The study area extent was drawn to include the two feeder lakes identified by Kingslake et al. (2015). In hindsight, we should have extended it further back to consider other feeder lakes, but we were trying to keep the study area relatively small to reduce computer processing time. We are unfortunately unable to change this now as the full algorithm would have to be re-run. Despite this, we think the overall results and conclusions of the paper would remain largely unchanged if we did extend the study area further.

Line 839: Are the white stripes across the figure artefacts? Consider changing the colour ramp to make values easier to read, and outlining lakes in black for visibility.

We have modified the colour ramp. The white stripes were an odd artefact during image export, and have been removed. Unfortunately, we could not outline the lakes in black due to the format of the data. Please note this figure is now Figure 4.

Line 852: Again, the colours make visibility difficult. I struggle to pick out areas in the 'always a stream' category

Colour scheme has been amended (Fig 6)

Line 877: I find this figure slightly confusing, though perhaps I have mis-interpreted it – why is one bar not shown per individual date studied in the 2016-2017 melt season (i.e. only 8 bars are shown, not 11). Clarify caption to show that loss events record either lake drainage or freeze- through.

We only show 8 bars as no loss events were recorded on the other 3 dates. We have added the following statement to the caption (Figure 10): "A loss event is defined as a > 80 % loss in water body volume through either lake drainage or freeze-through."

**Technical corrections**

Line 23: inconsistent hyphenation of Landsat-8 and Sentinel-2 throughout the manuscript; please check throughout for consistency

Landsat 8 should not be hyphenated, whilst Sentinel-2 should be, we have checked this throughout the document.

Line 68: inconsistent hyphenation of sub-surface; please check throughout for consistency

Corrected

Line 69: re-order citations chronologically; please check and modify throughout

Corrected

Line 74: delete 'events'

Corrected (Line 83)

Line 76: delete comma after 'front'

Corrected (Line 84)

Line 81: change 'on' to 'in'

No longer necessary as wording altered to improve the flow of the text.

*Line 102: is 'approximately' 123 km wide by 92 km long.*

Corrected using '~' (line 139).

*Line 161: 'change 'in' to 'into'*

Corrected (Line 213)

*Line 175: Change 'give' to 'produce'*

Corrected (Line 226)

*Line 201: Hyphenate 'water covered'*

We have since removed this sentence.

*Line 218: Delete 'different'*

Corrected (Line 270), start of paragraph shortened.

*Line 206: Change 'as' to 'because'*

Corrected (Line 258)

*Line 220: Correct (Philpot, 1989) to Philpot (1989)*

Corrected (Line 272)

*Line 294: Change 'track' to 'tracked'*

Wording changed (line 511).

*Line 393: Maximum should be ~3 $x10^6$ $m^2$, not 2.5 $x10^7$?*

We are not sure where this confusion has come from, perhaps you were looking at the envelopment transitions in Figure 6 (now Fig. 7)? This is likely because we were not using an appropriate figure for Fig.7 (now Fig.8), which we have now changed based on your previous recommendations.

*Line 426: change 'by' to 'on' (as this is showing loss events on that particular date rather than cumulative events, if I have understood correctly)*

*'By' is an appropriate word choice here, as the loss events did not occur on the single date, but rather 'by' this date.*

*Line 449: Change 'weather' to 'temperature fluctuations'*

Corrected (Line 549)

*Line 467: New sentence after 'bodies'. 'We suggest this occurs when water ponds...'*

This is no longer necessary as the text in this paragraph has been further edited.

*Line 471: New sentence after 'dam'. 'Therefore, it is more likely to be the result...'*

Corrected, with slightly different wording (Line 573).

*Line 497: hyphenate snow/firn-covered*

Corrected (Line 616)

*Line 500: add 'firn' into 'available pore space'*

Wording since altered .

*Line 547: hyphenate 'large scale'*

Corrected (throughout text)

*Line 863: Western and Eastern System are the wrong way round in the figure caption. Wrong dates in legends?*

Based on your recommendations (above) we have removed this figure and the text relating to it, replacing it with a more suitable figure showing the volume and area of the ES and WS (Fig. 8).

**Response to Reviewer Comments – Reviewer 2**

*This manuscript describes the evolution of surface meltwater on the Nivlisen Ice Shelf in Antarctica during the 2016-2017 melt season. The authors describe in detail their adaptation of the FAST algorithm for tracking water on ice shelves, and apply the method to two optical satellite datasets. They develop four categories of ice-shelf water bodies, and use them to discuss water transport across the Nivlisen Ice Shelf. Two main systems emerge on the ice shelf along linear surface depressions as the surface hydrology system transitions from isolated lakes to a connected network.*

*As explained in the introduction, ice-shelf surface hydrology is a critical component of ice-shelf stability, and thus important in predictions of ice-sheet mass loss. Being able to map and analyse the development of ice-shelf surface hydrology is important for understanding how future warming and subsequent ice-shelf melting will influence ice shelf behaviour. There is a significant gap in our knowledge of water transport across ice shelves, and the goals of this manuscript are both timely and within the scope of TC. In general this manuscript is well written and should be published in TC with some revisions applied. The topic of ice-shelf meltwater transport is important, and the results of this study will certainly be of interest to the TC audience.*

We thank Reviewer 2 for their positive and constructive overview and further feedback on this paper.

**Major Comments:**

*This manuscript builds on previously developed tools (water-tracking algorithms, band ratios, etc) and applies them to an ice-shelf whose surface hydrology has not to my knowledge been analysed in detail. The authors have adapted these methods and tools to suit ice shelves, and combined this algorithm with previously validated methods of water volume retrieval from satellite imagery. However, the authors do not clarify why the FAST algorithm is not suitable for use on ice shelves and do not explain the differences between the FASTER algorithm and their new FASTISh.*

Additional major comment: *In terms of the methods, providing context for the FASTISh algorithm would go a long way. A lot of time is spent in the beginning discussing a method that is at this point standard in ice-shelf/ice-sheet hydrology, and it's not clear how FASTISh differs from FAST.*

Additional minor comment: *Line 92: Why is it necessary to produce FASTISh? It would be very helpful to include a description of FAST, highlighting its strengths and weaknesses.*

We have provided justifications for adapting FASTER for application to Antarctic ice shelves in lines 100-117, we have also made the differences between the FASTER and FASTISh algorithms clearer (lines 121-133).

*The authors clearly demonstrate an evolving meltwater system on the Nivlisen Ice Shelf, and clearly describe why these observations are important for ice-shelf*

*stability. However, the conclusions could use some clarification and possibly additional analysis. First of all, the difference between a meltwater lake and a stream is defined here only geometrically. Streams and rivers are defined by water flow, but the definition used here does not include a water flow condition. Rivers and streams described here could technically be linear lakes that fill up from percolation through the firn. The authors seem to support this, since they describe two mechanisms for water transport 1) movement through firn and 2) movement via overtopping lakes, but then claim there is no evidence of overtopping lakes. There is not sufficient data or analysis to confirm water is flowing in these features precisely because water transport in the firn cannot be tracked from optical satellite imagery.*

*And a minor comment: Line 357-358: I am now getting confused between lakes and streams- are the two "large streams" just linear lakes? Your distinction between the two is only geometric. It is not clear that water flows in these water bodies the way water flows in streams. I am convinced water is transported, but I don't think there is clear evidence for how. Except that you later say you have no evidence for water overtopping lakes, which makes me think you are concluding most water moves through firn.*

Based on our current level of analysis, we now agree that it is not possible to confidently categorise water bodies as lakes or streams. We have therefore re-named our categories to purely reflect the geometry of the water bodies, and water bodies are classified as either circular or linear. These adjustments have been made where appropriate throughout the manuscript. We also argue throughout the discussion for the lateral movement of water through the firn pack (e.g lines 583-597).

We have further clarified our argument for overtopping of lakes in the discussion. We DO argue for potential overtopping of water bodies (see lines 568-571), but we argue that the two large linear systems (the ES and the WS) do not develop through overtopping a dam, rather they just follow their downward sloping profiles as the melt season progresses (see lines 571-575 and 556-558).

*Satellite radar could be used to track melt in the upper firn layers and might provide insights into exactly how water is moving. Additionally, the authors have not explained where firn is located on the ice shelf or how it is detected. I think it was visually identified. That should be briefly described and indicated on one or more maps (along with blue ice areas).*

We have incorporated a Sentinel-1 SAR image into our discussion and into a new figure (Figure 11). We have used this to further our argument for lateral water transfer across the ice shelf through the ES and WS. This figure also provides evidence for both the ES and WS infiltrating into the firn pack (lines 583-597). We have also added a description of where we believe the firn to be located in Section 2 (lines 145-147).

*The connection between temperature and surface hydrology development could be enhanced by including data from a climate model that covers the entire ice shelf as opposed to a single weather station. The development of air and ground*

*temperatures could be tracked through space and time and then be compared to the development of the surface hydrology system that the authors already quantified.*

And: *Also I think a single point of temperature is insufficient for this analysis. It would be useful to see modeled (maybe RACMO) temperature and energy balance across the extent of the surface hydrology system.*

We agree with this, and we have therefore analysed air temperatures "from an atmosphere-only regional climate CORDEX (COordinated Regional climate Downscaling Experiment) simulation of Antarctica using the limited-area configuration of Version 11.1 of the UK Met Office Unified Model (MetUM) for the period 2016-2017." Please see section 3.7 for a full explanation.

We have, however, not looked at spatial gradients in the climate data, as we believe this is something that is beyond the scope of this current study.

*The general structure of the manuscript could be improved by including some of the more important conclusions and implications in the abstract and introduction. For example, the comparison to the surface lakes of Larsen B in the discussion could be highlighted to provide some motivation for the study, as opposed to limiting the goal of the study to developing an algorithm and tracking water on one ice shelf.*

We agree with this comment and have made adjustments in the abstract (lines 29-33), and introduction (lines 63-67).

*Some more context for the Nivlisen Ice Shelf would be helpful in Section 2.*

We have added more context for the Nivlisen Ice Shelf, and have described the blue ice extent and firn extent in greater detail (lines 145-147), we have also expanded our justifications for selecting Nivlisen Ice Shelf as the study area (lines 150-157).

**Figures:**

*Titles would help the reader interpret figures more easily.*

Corrected throughout

*Figure 1. Add coordinates to Nivlisen map, include location of weather station. Annotate blue ice areas, water bodies and firn extent. The caption refers to a red star to locate the Nivlisen Ice Shelf, but I don't see it. Also, is this image mosaicked? From the description of data preparation, it seems like the entire ice shelf cannot be captured by a single Landsat scene, and that pairs of scenes were mosaicked.*

We have added blue ice areas and co-ordinates to Figure 1. To clarify the firn extent on the ice shelf we have added this statement to Section 2, as it would be clearer than labelling the map (lines 145-147): "Beyond this blue ice region, towards the north, the ice shelf transitions into an accumulation zone as the firn layer thickens (Horwath et al., 2006)."

The weather station data is no longer used for this study, so we have not added this location. Given the amount of information already on the figure, we have decided not to label the water bodies, as it would be difficult to label the full extent of all water here clearly

We have also clarified that the image is mosaicked (line 1000).

*Figure 2. This is a nice workflow diagram. One very detail-oriented comment: It is a little confusing to have "Cloud Mask" in an input, and then the next line be the "Apply Cloud Mask" processes in the same workflow. From my understanding of the workflow, cloud masks are generated from a separate workflow followed in Google Earth Engine and are then an output of that workflow. Maybe delete the "Cloud Mask" input from the top.*

Corrected (Fig.2).

*Figure 3. Include the paths of the DEM that were extracted and shown in Figure 8. Consider combining this figure with Figure 8 and maybe Figure 7. It would be nice to have an "elevation figure" to synthesize everything.*

We have removed Figure 7 based on recommendations made by Reviewer 1, and replaced it with a different Figure (now Figure 8). We would also like to keep Figures 3 (now Fig.4) and 8 (now Fig.9) separate, as they draw upon two different types of REMA data. Figure 3 (4) shows the 8 m mosaicked data product, whilst Figure 8 (9) utilises the 2 m data strips. The paths extracted for the DEM in Figure 8 are shown in Supplementary Figure 5.

*Figure 4. This shows the water transport really nicely. Consider having the scale bar on only one image so that the eye can focus more on the water. Also, consider making the color of lakes and streams more distinct- it is hard to distinguish in print and only slightly easier on screen. Also, the text references streams a and b on line 343, but I think you should keep names and labels consistent. Label them here, since it is the first time we see the hydrologic system. Maybe add labels to row D.*

All suggested changes made. However, we have continued to label the 'streams' (now called linear water bodies) as 'a' and 'b' as at this point in the results, as we have not introduced the ES and WS, which are products of the tracking results (which are not introduced until section 4.2). Please note this is now Figure 5.

*Figure 5. Great figure! Move the WS and ES labels so they don't overlap the nice masks. Is the basemap Landsat? Include that in the caption. Is it necessary to include a category for water that isn't mapped if the background shows a satellite image? It makes me want to look for what features with a gray outline.*

We have moved the labels, and changed the colours for each category to make the figure clearer and to reflect the changes made to Figure 5. We have also removed the category for water that isn't mapped, and have clarified where the base image is

from in the caption: "Base image aquired by Sentinel-2 on 26th January 2017" (Line 1049). Please note this is now Figure 6.

*Figure 6. Get rid of white space after 01-04-17 since nothing is plotted there.*

Corrected. Please note this is now Fig.7. We have also adjusted the line colours to reflect the changes made in Figures 4 (now Fig 5) and 5 (now Fig.6).

*Actually, if I'm correct in this interpretation, it seems like this figure shows a period of water deepening– in the All Water Bodies and Envelopment Transitions categories, the volume increases at a greater rate than the area from ~ 01-12-16 to 01-02-17.*

This is roughly correct, we have added the following statement to section 4.2.1 (lines 480-486): "Between 17th December 2016 and 27th December 2016 'all water bodies' are characterised by 'deepening', as their total volume increases at a greater rate than their total area, and their mean depth increases (Tables 1 and 2, Fig. 7). Whereas between the 27th December and the 26th January, 'all water bodies' are characterised by 'spreading', as their total area increases at a faster rate than their total volume, and the mean water body depth decreases (Tables 1 and 2, Fig. 7)."

*Figure 7. This figure basically tells me that water flows downhill. This is very encouraging, but I think that's communicated in the combination of Figures 3 and 4.*

We have removed this figure and replaced with another figure to address a comment from Reviewer 1 (please note this is now Figure 8).

**Specific comments:**

*Line 1: change "stored" to "can form"*

Wording adapted further to: "Surface meltwater on ice shelves can exist as slush, it can pond in lakes or crevasses, or it can flow in surface streams and rivers." (Lines 15-16).

*Line 31: I don't think it's clear that water transfer is "facilitated by two large surface streams." In fact, the discussion seems to indicate lateral transport of water through firn.*

We have reworded this to: "At this time, 63% of the total volume is held within two linear surface meltwater systems, which are up to 27 km long, are orientated along the ice shelf's north-south axis, and follow the surface slope. Over the course of the melt season, they appear to migrate away from the grounding line, while growing in size and enveloping smaller water bodies. This suggests there is large-scale lateral water transfer through the surface meltwater system and the firn pack towards the ice-shelf front during the summer." (Lines 28-33).

*Line 80: I agree it is important to assess the variability of ice-shelf surface hydrology over the course of a melt season, but the year-to-year change is also critical. As just*

*one example, the Nansen River does not form every year (Bell et al. 2017). If meltwater transport is such an important part of ice-shelf stability, then its variable behaviour must be important, too. What if water didn't move across Nivlisen one year?*

We have appended 'and across multiple melt seasons' to this sentence (line 90).

*Line 98: In general I would like more description of the ice shelf here. A discussion of the firn would be useful.*

We have added much more detail to section 2 (lines 137-157), including a description of the firn extent (lines 145-147).

*Line 103: Be more specific about where the ice thickness reaches 700 m. Is 700 m a grounding line thickness? Taking a look at Horwath et al., referred to in the text, it looks like it might be.*

Yes, it is towards the grounding line. Clarified on lines 139-141: "Ice thickness ranges from 150 m at the calving front to ~ 700 m towards the ice shelf's grounding line in the southeast, and it exhibits flow velocities of around 20 m a$^{-1}$ to 130 m a$^{-1}$ (Horwath et al., 2006)."

*Line 105: It would be helpful to include a reference to Fig. 1 here with some annotations provided.*

Corrected (Line 145)

*Line 110 (Study Area in general): Discuss meltwater features in more detail than "meltwater features show changes over time." Kinglsake et al. 2015 specifically say that lakes drain on the Nivlisen Ice Shelf. You may disagree with the classification of the meltwater identified by Kingslake et al. 2015 – the images there look like what you call an "envelope transition." However, "changes over time" doesn't communicate enough information to the reader. This description could include basic information from your analysis, too. Knowing up front how large the extent of melting is compared to the total area of the ice shelf, for example, would be helpful in orienting the reader. Also including some basic climate information would be good, especially since you talk about temperature data later.*

We have discussed the findings of Kingslake et al. (2015) in more detail (lines 153-155): "these meltwater features have shown significant development over a melt season, as source lakes upstream of the grounding line appeared to drain laterally, rapidly flooding large areas of the ice shelf  (Kingslake et al., 2015)"

We have also incorporated basic climate and melt extent information in lines 147-149: "In the 2016-2017 melt season, air temperatures on the Nivlisen Ice Shelf ranged between ~ -25°C and 2°C, and 1.6 % of the study area occupied by a surface water body at least once during this time."

*Line 139: Converting from DN to TOA values is important for analysing spectra anywhere, not just at high latitudes.*

This is true, but typically landsat scenes are corrected for TOA using a single solar angle over the full image scene, here, we correct each pixel with individual solar angles, as at high latitudes the solar angle can vary significantly over a full image scene. We have clarified this within the text (lines 179-187):

"Typically, Landsat scenes are converted to TOA reflectance values using a single solar angle over the whole image scene. However, here we correct each pixel for the specific solar illumination angle, based on metadata stored in the .ANG file, and using the 'Solar and View Angle Generation Algorithm' provided by NASA (https://landsat.usgs.gov/sites/default/files/documents/LSDS-1928_L8-OLI-TIRS_Solar-View-Angle-Generation_ADD.pdf). Converting from DN to TOA values on a per-pixel basis is imperative when mosaicking and comparing images obtained at high latitudes, as the solar angle at the time of acquisition can vary significantly across each scene due to the large change in longitude."

*Line 156: I appreciate that clipping each scene to the same extent is a required step in the algorithm, but I don't understand the reason. Although you say "when comparing images," but you can compare scenes that have overlapping areas but not the same extent. This makes me think there is some other requirement that you haven't mentioned.*

We have clarified this in lines 206-209:

"Clipping each scene to the same extent is required when comparing images through the FASTISh algorithm, as tracking individual features over time requires images with a consistent spatial reference frame to determine the location of each water body."

*Line 168: It might be good to describe bands in terms of their spectral range rather than colour, since you are using imagery from two different platforms. Clearly this is not critical, just a thought.*

For simplicity for the reader and easy comparison between the two platforms, we would like to keep these as colours.

*Line 171: The division by a "quantification value" comes out of nowhere. Is there a paper you could cite that explains this as a part of data processing? I would delete "expected" since it can imply that something is wrong with Sentinel-2, when it seems like it's just a convention of how they deliver their data. I haven't worked much with Sentinel-2, however, so if this really is an unexpected discrepancy then ignore my suggestion.*

We have deleted 'expected' as suggested (line 222), and have included a supporting citation (Traganos et al., 2018) (line 223-224).

*Line 178: Is the thresholding approach the same as before?*

No, we have clarified this by adjusting our wording (Lines 229-232): "As the simple cloud score algorithm had not been adapted for application to S2 imagery at the time of writing, we computed a cloud mask for each image using a thresholding approach, whereby pixels were categorised as cloudy if the SWIR band value was > 10,000."

*Line 179: How can a value exceed 10,000 if all bands were divided by 10,000?*

We only converted bands 2,3,4 to the 0-1 range as these were used in main area and depth analysis. We did not convert band 11 as it seemed unnecessary. We have better clarified this in lines (224-226).

*Line 233: While I know it is correct to say that errors in lake depth estimates have been calculated as zero (Pope et al. 2016), I think it is misleading to have this be the only mention of error about this method. To a reader who isn't already familiar with this method, the claim that error is zero will be shocking, and probably lead to skepticism. I think that if you are going to describe the method to help the reader understand what you did, it is important to give a little bit more detail about the sensitivity to A and g, the dependence of the band chosen, and that physical qualities of the system are known (water column attenuation, surface roughess of the lake, etc.)*

We agree with this, and we received a very similar comment from Reviewer 1. We have incorporated a comprehensive review of the limitations and assumptions of the depth algorithm into Section 3 (lines 283-290).

*Line 265: I'd really like to see a figure of the thresholds and different water body shapes. I think it would help me understand the categories you introduce in section 3.5.*

We have included a new figure (Fig.3), showing three of the thresholds tested. This is clearer than showing every threshold tested.

*Line 286: There should definitely be a figure for this. It could be Figure 5, but I would almost prefer a zoomed-in high detail of an area with one of each type.*

We feel that Figure 5 (now Figure 6) really does do this, we have made the colour scheme clearer. The addition of the new Figure 3 will also help to show how we selected linear and circular water bodies, which feed into the final tracked water body categories.

*Line 308: Is the figure for this Fig. 7? If so, refer to it. But also, be clear about which water bodies. I was expecting an elevation profile with different colours representing your four water body categories, whereas Fig. 7 shows the Eastern and Western Systems.*

We have clarified this (lines 370-373): "In addition, a single 2 m REMA data strip from 31st January 2016 was used to extract the elevation profiles along two tracked water bodies, the Eastern System and the Western System, which are introduced in section

4.2.2 " However, we do not want to refer to the Figure (now Fig.9) at this point in the paper, as it will introduce results too early in the paper.

*Section 3.7: Plot this on a map- I can't figure out where it is!*

As we are now using modelled climate data rather than weather station data, this is no longer necessary.

*Line 326 and Section 4.1: Give a general synthesis. Instead of only having four paragraphs that describe the observations, tell the reader the overall trend. I can't really grab on to a take-away message the way it is written.*

We have synthesised the main message in an 'introductory' paragraph to this section (lines 414-418):

"The seasonal evolution of meltwater bodies during the 2016-2017 summer is shown in Figure 5. The surface meltwater system transitions from a series of small isolated water bodies clustered towards the grounding line (Fig 5A), to a connected system dominated by two linear water bodies with a length of (a) ~ 20.5 km and (b) ~ 16.9 km that propagate towards the ice-shelf front (Fig 5D)."

We have then improved the flow of the following text (lines 420-458).

*Line 328-29: Ice shelves are flat places- what does relatively flat mean and how do you know how thin the winter snow cover is?*

We have adjusted the text (Lines 420-422) to read as follows: On 11th December 2016, few meltwater bodies exist, and they are predominantly clustered within the blue ice region towards the grounding line in the south-west (Fig 5.A).

*Line 343-344: Refer to Fig. 5.*

We think it is actually better to refer to Figure 4 (now Figure 5) rather than Figure 5 (now Figure 6) here.

*Line 353: Again, how do you know the extent of the firn? If it is visually identified, consider digitizing it. I'd be curious to see how the firn extent changes as the system evolves, especially given your discussion section.*

The firn extent covers the ice shelf area north of the blue ice region. We have clearly labelled the blue ice region in Fig.1 and have also added a description of the firn extent into Section 2 (lines 145-147).

*Line 385-387: The FASTISh algorithm itself does not produce a time series of water body properties- it only does you have used it on a time series of images.*

We have reworded this sentence to: "In addition to quantifying total surface water area and volume for each of the four water body categories (Fig. 7, Table 2), the

FASTISh algorithm also tracks changes in the area and volume of *individual* water bodies." (Lines 490-492).

*Line 392-397: While you refer to Fig. 7 for this, that figure only shows migration to lower elevations, not lower latitudes. I am not sure what trends you refer to in section 4.1 – rewriting that section to include a general point would help. Then refer to it here by saying "In line with the XYZ trends described in section 4.1. . ..."*

We did not write this section well previously, we have since removed and replaced Figure 7 (now Figure 8) in response to a combination of suggestions from both reviewers. Figure 7 (now Figure 8) no longer shows the elevation profile of the ES and WS, but instead it shows the total area and volume of the ES and WS. This is described in lines 498-502. We then describe the surface slope followed by both systems (lines 502-507), with reference to Figure 9.

*Line 399: I got confused between the Eastern System and Western System and the water body categories. When I looked at Fig. 7 I was expecting to see the four categories, since the title of the section is "Tracking Individual Water Bodies." Why have you not done this? Is it because most of the water is in the ES and WS? It would make sense to focus on these two, given that there is also migration of both systems. But I think that needs to be clarified in the text. It is even more confusing in the rest of the paragraph because on line 404 you say "most of the surface water" and it's not clear if you mean "most of the surface water in each meltwater system" (and I don't even know which meltwater system it is, east or west?) or if you mean "most of the surface water on the ice shelf".*

Figure 6 (now Figure 7) shows the time series for the four distinct categories. But, yes, we focus Figure 7 (now Figure 8) on the ES and WS as they hold the majority of the surface melt at the peak of the melt season. We have clarified in that we are only referring to the ES and WS within section 4.2.2 (lines 495-496).

We no longer refer to 'most of the surface water' – as this section has been largely re-written to reflect the changes made to Fig.7 (now Fig.8).

*Line 418-419: I'm not sure this sentence matters. What is rhythmic variance? What do these properties mean for the surface water?*

We have removed this sentence.

*Line 448: You say surface water area and volume correspond with rising temperatures, but Fig. 9 shows volume loss is associated with temperature increase, and only in the category of "always a lake" when most of the water is contained in the ES and WS. It would be helpful to include a figure showing what is claimed on line 453 "As temperatures rise and surface water bodies increase in area and volume. . .." It seems like the point you are trying to make in this section is that temperature controls the evolution of the system, so showing the decreasing lakes with increasing envelopment transition might make that clearer. Although later it seems like you are claiming the opposite.*

And *Line 508-526: It seems like now you are saying temperature does not control the evolution of the surface hydrology system. Earlier it seemed like you were saying the opposite.*

We have modified Fig.9 (now Fig.10) into 2 panels. Panel 'a' shows the temperature data and panel 'b' shows the volume loss from circular water bodies as well as a plot of the total surface water volume observed on each date.

"The loss of $1.5 \times 10^7$ m$^3$ of surface water from the circular water bodies by 27th December 2017 follows sustained relatively warmer atmospheric conditions since the beginning of December 2017 (Fig. 10), and coincides with an increase in the total surface water volume on the ice shelf (Fig 10b). In particular, we see an increase in the volume of water held within the enveloping water bodies, which continues to increase up to a maximum of $4.5 \times 10^7$ m$^3$ on 26th January 2017 (Fig. 7). It is likely, therefore, that the loss of water from circular water bodies at this early stage in the melt season signifies the lateral transfer of water away from the small 'isolated' bodies near the grounding line into the large enveloping water bodies which hold and transport the surface meltwater across the ice shelf to more distal regions." (Lines 628-636).

*Line 463: How do you know water is flowing along linear depressions as opposed to those depressions being filled from the sides due to transport in the firn? Especially since you mention on line 470-472 that the growth of ES and WS are more likely to be fed by increased meltwater production and movement through firn. In general, I think you argue well for meltwater transport through firn. I would highlight that, but it would require a little more discussion and analysis of how you track firn on the surface.*

We have changed our statement (lines 563-564) to say: *"This is because these systems facilitate large-scale transfer of water across the shelf, as water ponds within linear depressions."*

We have clarified the extent of the firn in Section 2 of the paper.

*Line 464-468: You have not demonstrated that the ES and WS are "fed by smaller surface lakes and streams both above and below the grounding line."*

We agree, so have removed this statement.

*Line 470-472: Is it really possible to infer this when there are no images between Dec. 27th and Jan 26th? I assume that evidence of a "lip" would be the development of a stream forming out of a lake. How do you know this didn't happen? Also, refer to Fig. 4 here.*

*From looking at the DEM data available to use, we see no significant lip/ dam blocking the flow of water in either the ES or WS at these points.*

*Line 480-482: You claim it is possible that meltwater could be transported to the front of the ice shelf in the future. That would require water to organize in a river that can*

*flow off the ice-shelf edge (unless there is some meltwater evacuation through firn that I don't know about, but that's another can of worms). You could potentially demonstrate this by assuming at some point the firn will saturate and water will follow surface depressions (you have already argued that this is happening). You could perform a water routing analysis on the full DEM, or at least see if you can visually identify linear surface depressions out to the edge of the shelf.*

We have expanded our argument slightly within lines 603-605, which builds upon our previous discussion.

We agree that flow routing analysis would be useful, and we attempted to carry out this analysis prior to initial submission. Unfortunately, the REMA data contains too many significant elevation errors and data gaps, meaning that flow routing analysis is extremely hard. 'Tidying up' the REMA data would be a large project in itself, and is something we think is beyond the scope of this current study. Visually, we can't yet see linear surface depressions propagating out towards the ice shelf edge.

*Line 488-501: "The relatively extensive snow and firn cover . . .likely prevents the development of. . .meltwater. . .on the Larsen B." This is a very important distinction that I think you should highlight (maybe include it in the abstract and allude to it in the introduction?). This is definitely an important conclusion, because it defines a type of meltwater type and evolution that stands in contrast to the "ponds break ice shelves" style of meltwater without needing to remove water.*

We have altered and clarified our argument slightly here (lines 617-621): "The development of these large, linear water bodies is likely facilitated by topography, and allows the transfer of summer meltwater towards the ice-shelf front. This large scale lateral transfer of meltwater is further facilitated as the ES and WS develop over frozen meltwater paths from previous years (Kingslake et al., 2015)."

Please note we do still discuss the importance of the firn layer in lines 583-597 and in lines 573-575.

*Line 515-518: Earlier you said that the transfer of water was \*not\* due to overtopping of surface lakes.*

Not quite, we argue that the progression of the ES and WS did not result from the overtopping of a topographic lip. However, this statement only applies to the ES and WS. We have clarified this in lines 565-575.

*Line 530-552: What are the implications for the stability of Nivlisen? Could you tie your analysis more explicitly to our general understanding of ice-shelf stability? Your refreezing calculation is really interesting, but has nothing to do with the rest of the paper. It makes me again want to know more about firn on this ice shelf.*

Beyond our discussion within Section 5.3 on the significance of ponding vs rivers exporting water off the ice shelf edge, and latent heat release upon re-freezing, we cannot think of much we can add without incorporating further analysis beyond the scope of the paper.

*Line 558-576: I would lead with the conclusions you reach about the ice shelf, not with a description of your method, and also include a summary of the implications of your analysis.*

We've kept the general structure of the conclusions as we think it's good to briefly summarise the methodology as well as the results. But we are happy to restructure it if the editor/journal style requires us to.

*Line 570: Again, how do you know there is a thin snow band and firn pack?*

This was a generous assumption as we likely can't differentiate between the snow and the firn, we have therefore removed this statement.

**Technical comments**

*Line 20: hyphenate "ice-shelf"*
*Line 59: hyphenate "ice-shelf"*
*Line 76: hyphenate "ice-shelf"*
*Line 548: hyphenate "ice-shelf"*

Corrected where appropriate throughout text.

*Line 210: change "two" to "2" since it follows a mathematical symbol, even though it is a number less than ten.*

Corrected (Line 262)

*Line 294: "identified and tracked"*

Wording changed, no longer applicable.

*Line 306-307: Spell out acronyms when you use them for the first time REMA and DEM.*

Corrected (Lines 368-369)

**References**

Horwath, M., Dietrich, R., Baessler, M., Nixdorf, U., Steinhage, D., Fritzsche, D., Damm, V. and Reitmayr, G.: Nivlisen, an Antarctic ice shelf in Dronning Maud Land: Geodetic-glaciological results from a combined analysis of ice thickness, ice surface height and ice-flow observations, J. Glaciol., 52(176), 17–30, doi:10.3189/172756506781828953, 2006

---

## Author Comment (AC3) · 23 Apr 2020

Thank you for your positive and constructive review, we have attached our responses as a supplement.

Please also note the supplement to this comment:
https://www.the-cryosphere-discuss.net/tc-2019-316/tc-2019-316-AC3-supplement.pdf

---

## Editor Decision (ED1)

I would like to thank both reviewers for their thorough and constructive comments on this manuscript and also the authors for posting their response to the reviewers' comments.

Both reviews are positive, but they raise a number of issues that require clarification or revision to the original manuscript. In their response, the authors clearly outline the steps that they propose to take to address all the points raised by the reviewers. I therefore invite them to submit a revised version of their manuscript.

Kind regards,

Pippa Whitehouse

Associate Editor, The Cryosphere

---

## Author Response (AR2)

30th May 2020

Dear Pippa Whitehouse,

**RE: Response to Editor Comments**

Thank you for your positive comments and suggested changes to the manuscript. We have responded to your comments (please see red text below) and have edited our manuscript as appropriate. Please also find attached a revised manuscript with tracked changes. The line numbers we refer to below are taken from the tracked changes manuscript.

Thank you for taking your time to consider our revised manuscript, we hope it is now acceptable for publication.

Kind Regards,

Rebecca Dell and co-authors.
* * *
Editor comments I would like to thank the authors for robustly addressing all of the reviewers' comments.

In the revised manuscript you have carried out a number of revisions that have improved the quality and presentation of the article and I am happy that there are no further major issues to be addressed. I list below a number of very minor technical points. Once these are addressed, I would be happy to accept this article for publication.

Kind regards,

Pippa Whitehouse
* * *
Line 47-48: hyphen needed in 'sea-level rise'

Corrected (L47-48)

Line 88: there is an extra space after 'streams'

Corrected (L88)

Line 120: "We develop the FASTER algorithm..." rephrase to indicate that you adapt/update the FASTER algorithm

Corrected to 'adapt' (L122)

Line 129: needs another closing bracket

Corrected (L131)

Line 148: word missing?

Corrected - Inserted 'was' (L151)

Line 162: upon -> on

Corrected (L165)

Line 222: I see you've converted numbers less than 10 into text, but here I think values are best expressed as '0 to 1' (but check the guidelines)

Corrected (L226)

Line 228: S2 -> Sentinel-2

Corrected (L232)

Line 291-292: clarify that you are still talking about Landsat 8 here

Changed 'In our study' to 'For Landsat 8 images' (L297)

Line 312: odd phrasing "pixel with an ice cover" (and at the start of the sentence)

We have reworded this sentence to: 'Pixels in the lake masks that were filled (normally those with a floating ice cover, see section 3.2) were assigned the mean water depth of their respective water bodies.' (L317-319)

Line 383: is -> are

**Corrected (L395)**

Line 387: listing the period covered by the model is confusing because later in the sentence you state that the model is used to provide a continuous time series from November 2016 to April 2017 – perhaps delete "for the period 20151231T1200Z to 20171230T0000Z" or split the sentence in two, making it clear that you are extracting data from a longer run.

We have split the sentence into two (L398-401).

Line 392: why not extract data from the grid point immediately to the north of Schirmacheroasen, since this is where the lakes are?

Sorry, this is a typo, we did actually use the cell to the north! We have corrected this. (L403)

Line 466: it does not make sense to say that the volume increases at a greater rate than the area because these quantities have different units (similarly line 469).

We have changed this section to: 'Between 17th December 2016 and 27th December 2016 'all water bodies' are effectively deepening, as their mean depth increases whilst the total area increases,

whereas between the 27th December and the 26th January 'all water bodies' are effectively spreading, as their mean depth decreases whilst total area increases. (Table 1, Fig. 7).' (L479-483).

Line 490: figure 9b suggests the elevation of the ES decreases by ~7 m over 27 km

**Corrected (L517)**

Line 556: perhaps mark the downstream extent of the WS and ES at various dates on figure 9, and include a reference to this figure in the text

Corrected (Fig 9 and L585).

Line 570: clarify that you are talking about figure 11b when you mention 'dark areas'

**Corrected (L599).**

Line 570: there is an unexplained reference to 'tributary glaciers' on this line

We have removed this and changed the sentence to 'Areas of low backscatter (appearing as dark areas in Figure 11b) extend across the grounding line onto the upper part of the ice shelf.' (L597-600).

Line 572: where relevant, clarify what aspect of these factors would result in low backscatter, e.g. particularly large/small/variable snow-grain size?

We have clarified and included a new reference (Sun et al., 2015) (L600-602):

Whilst areas of low backscatter may result from relatively small dry-snow grain sizes, shallow drysnow depths to underlying rougher surfaces, high surface roughnesses, or complex internal stratigraphies (Rott and Mätzler, 1987; Sun et al., 2015).

Sun, S., Che, T., Wang, J., Li, H., Hao, X., Wang, Z. and Wang, J.: Estimation and analysis of snow water equivalents based on C-band SAR data and field measurements, Arctic, Antarct. Alp. Res., 47(2), 313–326, doi:10.1657/AAAR00C-13-135, 2015.

Line 577: there is no scale bar to allow the reader to identify the backscatter values

**Corrected (Fig.11)**

Line 602: topography -> ice shelf surface topography

**Corrected (L631)**

Line 604: do you know this, and can you make any comment about what happens to the water in the WS and ES at the end of the 2016-17 melt season?

We don't know for certain, but we think it is likely based on the work of Kingslake et al. (2015). We have added 'likely' into our sentence (L633): 'This large scale lateral transfer of meltwater is likely further facilitated as the ES and WS develop over frozen meltwater paths from previous years

(Kingslake et al., 2015). Without further data and analysis we cannot speculate further on what happens to the water at the end of the melt season.

Line 653: hyphen needed in 'meltwater-driven instabilities' ?

**Corrected (L687)**

Some text is green! Or blue.

**Corrected throughout.**

Fig. 1: I suggest labelling the ice shelf (or say that the orange shape roughly delineates the ice shelf).

We have adjusted the figure caption as follows: The solid orange line roughly delineates the ice shelf and shows the study area extent used for this study (L1032-2033)

Fig. 2: why are there dot-dash lines around some boxes? May need a little re-designing around the 'Average Water Body Depths' box because this step is not carried out for Sentinel-2 data.

The dot-dash lines indicate steps that were only applied to Landsat 8 images, this is explained in the caption (L1044-1046):

'Dashed lines indicate steps that were applied to Landsat 8 images only, whereas solid lines indicate steps that were applied to both sets of image types.'

Fig. 4: could refer to this figure in section 3.5, i.e. where you first mention the maximum extent mask.

**Corrected (L345).**

Fig. 5: mention the date stamp in the caption; scale bar text is very small.

We have added the following to the caption: '*Date stamps are in the bottom right hand corner of each image.*' We have also increased the font size for the scale bar. (L1070).

Fig. 5 caption (line 1020): there is reference to 'lake and stream' masks rather than circular and linear features

**Corrected (L1069).**

Fig. 10 caption (line 1058): space missing after 'Schirmacheroasen'

**Corrected (L1109).**

Fig. 10: 'total volume lost' is plotted as a volume on a specific date, but presumably the meltwater disappears sometime between the previous image and that date? This could be clearer in the caption. Also, if this is correct, then should the volume be scaled to reflect the time period over which the meltwater disappeared?

The 'total volume lost' plot is plotted on the end date by which the circular water body has lost > 80 % of its volume (we call this a loss event). The loss event could have started at *any* date before this,

not necessarily just at the previous time point. We have therefore not scaled the data to reflect this as we believe the plot would be too busy, and harder to interpret. It is simpler to convey that 'by X date, Y number of lakes have lost > 80 % of their volume, equating to Z m3 of water'. We have edited the caption to better explain the data:

b) The total volume lost in 'loss events' **by each image date** from water bodies in the 'always circular' category (blue bars) and the total combined water volume (blue line). A loss event is defined as a > 80 % loss in water body volume through either lake drainage or freeze-through. The total number of loss events for each date is indicated above each bar. (L1109-1113).

Table 2: terminology refers to 'lakes' and 'streams' rather than linear and circular water bodies

**Corrected**

I note a couple of very recent papers that are relevant to this study. One is just published, and one has just come online in The Cryosphere Discussions (and I'm not sure whether you would be allowed to cite it, but I thought it worth mentioning). Entirely up to you as to whether you feel it would be useful to include them:

Arthur, J.F., Stokes, C.R., Jamieson, S.S.R., Carr, J.R. & Leeson, A.A. (2020). Recent understanding of Antarctic supraglacial lakes using satellite remote sensing. *Progress in Physical Geography*

**Included (L80)**

Fair, Z., Flanner, M., Brunt, K. M., Fricker, H. A., and Gardner, A. S.: Using ICESat-2 and Operation IceBridge altimetry for supraglacial lake depth retrievals, *The Cryosphere Discuss.*, https://doi.org/10.5194/tc-2020-136, in review, 2020. Included (Line 712-713). Also added an additional reference here too:

[revised manuscript text omitted]

---

## Editor Decision (ED2)

I would like to thank the authors for robustly addressing all of the reviewers' comments.

In the revised manuscript you have carried out a number of revisions that have improved the quality and presentation of the article and I am happy that there are no further major issues to be addressed. I list below a number of very minor technical points. Once these are addressed, I would be happy to accept this article for publication.

Kind regards,

Pippa Whitehouse
* * *
Line 47-48: hyphen needed in 'sea-level rise'

Line 88: there is an extra space after 'streams'

Line 120: "We develop the FASTER algorithm…" rephrase to indicate that you adapt/update the FASTER algorithm

Line 129: needs another closing bracket

Line 148: word missing?

Line 162: upon -> on

Line 222: I see you've converted numbers less than 10 into text, but here I think values are best expressed as '0 to 1' (but check the guidelines)

Line 228: S2 -> Sentinel-2

Line 291-292: clarify that you are still talking about Landsat 8 here

Line 312: odd phrasing "pixel with an ice cover" (and at the start of the sentence)

Line 383: is -> are

Line 387: listing the period covered by the model is confusing because later in the sentence you state that the model is used to provide a continuous time series from November 2016 to April 2017 – perhaps delete "for the period 20151231T1200Z to 20171230T0000Z" or split the sentence in two, making it clear that you are extracting data from a longer run

Line 392: why not extract data from the grid point immediately to the north of Schirmacheroasen, since this is where the lakes are?

Line 466: it does not make sense to say that the volume increases at a greater rate than the area because these quantities have different units (similarly line 469)

Line 490: figure 9b suggests the elevation of the ES decreases by ~7 m over 27 km

Line 556: perhaps mark the downstream extent of the WS and ES at various dates on figure 9, and include a reference to this figure in the text

Line 570: clarify that you are talking about figure 11b when you mention 'dark areas'

Line 570: there is an unexplained reference to 'tributary glaciers' on this line

Line 572: where relevant, clarify what aspect of these factors would result in low backscatter, e.g. particularly large/small/variable snow-grain size?

Line 577: there is no scale bar to allow the reader to identify the backscatter values

Line 602: topography -> ice shelf surface topography

Line 604: do you know this, and can you make any comment about what happens to the water in the WS and ES at the end of the 2016-17 melt season?

Line 653: hyphen needed in 'meltwater-driven instabilities' ?

Some text is green! Or blue.

Fig. 1: I suggest labelling the ice shelf (or say that the orange shape roughly delineates the ice shelf)

Fig. 2: why are there dot-dash lines around some boxes? May need a little re-designing around the 'Average Water Body Depths' box because this step is not carried out for Sentinel-2 data

Fig. 4: could refer to this figure in section 3.5, i.e. where you first mention the maximum extent mask

Fig. 5: mention the date stamp in the caption; scale bar text is very small

Fig. 5 caption (line 1020): there is reference to 'lake and stream' masks rather than circular and linear features

Fig. 10 caption (line 1058): space missing after 'Schirmacheroasen'

Fig. 10: 'total volume lost' is plotted as a volume on a specific date, but presumably the meltwater disappears sometime between the previous image and that date? This could be clearer in the caption. Also, if this is correct, then should the volume be scaled to reflect the time period over which the meltwater disappeared?

Table 2: terminology refers to 'lakes' and 'streams' rather than linear and circular water bodies
* * *
I note a couple of very recent papers that are relevant to this study. One is just published, and one has just come online in The Cryosphere Discussions (and I'm not sure whether you would be allowed to cite it, but I thought it worth mentioning). Entirely up to you as to whether you feel it would be useful to include them:

Arthur, J.F., Stokes, C.R., Jamieson, S.S.R., Carr, J.R. & Leeson, A.A. (2020). Recent understanding of Antarctic supraglacial lakes using satellite remote sensing. *Progress in Physical Geography*

Fair, Z., Flanner, M., Brunt, K. M., Fricker, H. A., and Gardner, A. S.: Using ICESat-2 and Operation IceBridge altimetry for supraglacial lake depth retrievals, *The Cryosphere Discuss*., https://doi.org/10.5194/tc-2020-136, in review, 2020.